# Collembola from the Gypsum Karst of Sorbas (Almería, Spain), with Descriptions of Three New Species

**DOI:** 10.3390/insects16030309

**Published:** 2025-03-16

**Authors:** Enrique Baquero, Pablo Barranco, Rafael Jordana

**Affiliations:** 1Institute for Biodiversity and Environment BIOMA, University of Navarra, 31008 Pamplona, Spain; 2Department of Environmental Biology, School of Sciences, University of Navarra, University Campus, 31008 Pamplona, Spain; rjordana@unav.es; 3Department of Biology and Geology, University of Almería, Centro de Investigación de Colecciones Científicas de la Universidad de Almería CECOUAL, Cite-IIB, La Cañada, 04120 Almería, Spain; pbvega@ual.es

**Keywords:** springtails, new species, gypsum caves, cave fauna

## Abstract

The Yesos de Sorbas cave complex (Almería, Spain) hosts a rich diversity of Collembola, small arthropods essential for ecosystem function. Over seven years, we conducted 83 visits, collecting 7875 specimens from four orders. The most abundant groups were Symphypleona and Entomobryomorpha, representing nine families. A total of 25 species were identified, including 21 known and 4 newly described ones. This study, based on microscopic morphological analysis, enhances taxonomic knowledge and cave ecology. The fragility of gypsum caves highlights the need for conservation efforts. It also underscores caves as biodiversity reservoirs, sheltering unique and sometimes undiscovered species. Ongoing biological exploration in these environments may reveal additional species, further enriching our understanding of subterranean life.

## 1. Introduction

The gypsum karst of Sorbas (GKS) (Almería, SE Spain) was declared a protected area in 1989 [1]. It is one of the best-known karst areas in Spain. More than one thousand caves and sinkholes are concentrated in only 25 km^2^, forming one of the world’s most important gypsum karst areas from a cave science perspective. GKS is located in the Tabernas–Sorbas Basin, SE Spain, and its climate is semiarid, with a mean annual precipitation of around 300 mm. The subterranean network, extending tens of kilometres beneath the arid, gypsiferous desert, contains the largest karstic complex in Andalusia [2]. Up to six passage levels have been detected in the gypsum caves’ development using the stratification planes between the soft marls and the more soluble gypsum units. This genetic duality indicates that the GKS can be considered an example of interstratal karstification. Most of the cave passages in this karst show a predominantly horizontal attitude and have relatively low vertical development (less than 10 m in many cases) and multiple entrances [3]. Within the GKS, the Cueva del Agua is the most extensive in Andalusia and the biggest gypsum cave in the whole of Spain. The known galleries extend over more than 8.9 km in the interior of the gypsum. More than one hundred dolines have been mapped in an area covering 1 km^2^ in the surroundings of the Cueva del Agua, representing the highest density of dolines reported in Spain [4]. Among the labyrinthine galleries, there runs an underground river that is always flowing, ending at the emergence of the spring. The galleries are very diverse; some of them have a vadose origin (generated above the water level once the galleries had been abandoned by the constant water flow), and others have a phreatic origin (they formed below the water level once galleries were flooded) [3]. The air circulation inside this cave is complex and driven by a thermal disequilibrium between the outside air and in-cave air and water temperatures. The annual mean temperature in the main gallery oscillates around 15–7 °C, whereas the average water temperature is 14.5 °C [5].

This work is part of the project CICYT-FEDER 1FD97-1577; it was carried out with the aim of sampling and studying the cave entomofauna in eight different cavities in the GKS. Partial results for the material studied have led to the publication of several studies on different taxonomic groups. Among them, three dealt with Coleoptera, two analyzed the spatio-temporal distribution in different cavities of Ptininae [6], and one explored *Trechus* species [7]. Others include the description of a new tenebrionid beetle [8], and one new pselaphid species is pending description [9]. Ten spider species have been reported to date, two of which constitute new taxa [10]. Two species of pseudoscorpions have been highlighted, one of which has been described as a new species [11]. A new species of silverfish of the genus *Coletinia* has also been described [12]. Finally, several citations have also been made, such as fleas [13].

## 2. Materials and Methods

### 2.1. Caves Studied

#### Caves Without Permanent Water Regimens

1. Covadura, 37.11881, −2.08285 (37°07′07.7″ N 2°04′58.3″ W) (WGS84), 355 m a.s.l. This is located in the northwestern part of the gypsum karst. This cave system comprises 4.25 km of galleries, distributed across six levels of depth, reaching −126 m deep. The air temperature fluctuates along the cave; the deepest galleries are 3 °C cooler than the upper ones. The cave temperature also varies within and across years, ranging from 9.2–11.8 °C in February to 14.2–16 °C in September. However, earlier microclimatic studies suggest the possible existence of hidden air entrances, with a strong influence on air movements in the cave [14]. It is a cave with humidity close to saturation, especially at the lower levels. This causes a high level of water condensation on the walls, ceiling, and fallen stones. The trophic contribution from the outside occurs through runoff, as well as guano from the colony of bats in the cavity. Sixteen sampling points were placed in this cave, mainly on the upper level. See Figure 1 for all the caves.

The Cueva C3 is located in the northern part of the gypsum karst. Its entrance lies near the western access point to Covadura Cave. This cave is developed in the upper marl level of the Yesares formation, around 3 m deep, and is 150 m long. The height of its passages ranges between 1.5 and 3 m [15]. The median cave air temperature is 16.2 °C; it fluctuates between 16.2 and 17.8 °C. Six sampling points were placed in this cave.

2. Sima del Camión, 37.12366, −2.06566 (37°07′25.2″ N 2°03′56.4″ W), 418 m a.s.l. It is only 469 m long and reaches a depth of −60 m. It is a seasonal watercourse, although there are continuous pools on the loams that humidify the environment. It has two main levels with several inlets. Organic matter is brought in by material being dragged through small gullies that connect with the exterior. Seven sampling points were placed in this cave.

3. Complejo GEP (GEP Complex), 37.11465, −2.06577 (37°06′52.7″ N 2°03′56.8″ W), 380 m a.s.l. It has a length of 1080 m and a maximum depth of −60 m. The most important characteristic of this cavity is that, throughout its development, it combines rooms of xeric environments with others with a higher degree of humidity. The cave temperature varies between chambers, from a minimal 12.6–14.9 °C to a maximal 13.7–16.2 °C. It receives an important trophic contribution through the fissures that connect with the exterior in certain areas. Twenty-two sampling points were placed in this cave.

4. Sima los Pinos, 37.12139, −2.05555 (37°07′17.0″ N 2°03′20.0″ W), 400 m a.s.l. It has a short length of only 240 m, and its depth is −60 m. It is a cavity with high humidity due to the continuous supply of infiltrated water and a permanent watercourse in the deepest part of the cavity. Six sampling points were placed on the upper level of this cave.

#### Caves with Constant Water Regimens and Which Were Colder Than the Previous Group Included the Following

5. Cueva del Tesoro, 37.095280, −2.070650 (37°05′43.0″ N 2°04′14.3″ W), 395 m a.s.l. It is located in a small gully that flows into the Barranco del Tesoro and has a total length of 4245 m. It is a cavity related to the sporadic water flow of a small gully with numerous connections with the exterior, which means that it is subject to constant airflow. Seventeen sampling points were placed throughout the cavity. Sampling was carried out during the same period as in the previous cave. Some traps were lost for various reasons, as previously mentioned. Seventeen sampling points were placed in this cave.

6. Cueva de los Apas (APAS Cave), 37.09662, −2.06599 (37°05′47.8″ N 2°03′57.6″ W), 372 m a.s.l. It is located in Barranco del Tesoro and has a more or less horizontal disposition, and the whole galleries extend over 1500 m. It is characterized by the presence of a permanent watercourse (constant regimen) and a dry upper zone, with low relative humidity. On this occasion, the sampling area was limited to the area adjacent to the watercourse due to the difficulty of placing traps in the dry zone. Six sampling points were placed. A large number of traps were lost for various reasons during the sampling period.

7. Cueva del Agua, 37.10546, −2.04337 (37°06′19.7″ N 2°02′36.1″ W), 360 m a.s.l. It comprises a system with more than 33 entrances, reaching 50 m deep, and its galleries extend over more than 8.9 km in the interior of the gypsum. The median cave air temperature is 16.1 °C. Like the previous example, it has a permanent watercourse with a constant regimen. A total of 21 sampling points were placed in this cave.

### 2.2. Sampling

The material studied was collected using pitfall traps. The number of traps per cave depended on the cave size; the traps were placed at the same points in each of the samples. The traps were placed in the total darkness zones of the caves; only in the GEP Complex and C-3 were 2 and 1 traps placed in the twilight zone, respectively. The distance between the traps varied, as these caves are made up of several rooms connected by galleries, with varying widths, and crawl spaces, which in some cases are made up entirely of gypsum crystals, making it difficult to excavate them in order to place a trap or making it impossible to place traps.

The traps were baited with sobrasada and filled with 50 mL of Turquin’s solution (1000 mL of beer, 10 g of chloral hydrate, 5 mL of acetic acid, and 2 mL of formalin) as an attractant in the Cueva del Yeso. We changed to a solution composed of beer, vinegar, and glycerine in the Cueva del Tesoro and in the Cueva de los Apas due to the abundant damage suffered in both caves due to small mammals (rodents). Sampling was performed bimonthly, with the traps exposed for one month.

All three caves are visited by tourists and cavers alike, although the Cueva de los Apas receives comparatively fewer visitors.

Some of the traps in certain samples could not be taken into account for various reasons: manipulation by visitors to the caves (they even threw coins in!), evaporation, being washed away by rainwater, and consumption by or deterioration due to rodents or mustelids.

The sampling period covered seven years, from 1996 (one sampling in 1996 and another in 1997) to 2003. After 83 visits to the caves, no two caves were visited on the same day (with one exception), and the samplings in each cave were as follows: 6 at C-3; 22 at the GEP Complex and Covadura; 5 at Apas Cave; 9 at Cueva del Agua; 15 at Cueva del Tesoro; and 4 at Sima del Camión. The number of specimens captured per day and cave ranged between 50 and 140, with the following results per cave: C-3, 111; GEP Complex, 140; Covadura, 72; Apas Cave, 68; Cueva del Agua, 92; Cueva del Tesoro, 76; and Sima del Camión, 49.

### 2.3. Material Processing

After preliminary sorting to separate the Symphypleona and Entomobryomorpha from other Collembola, some representative specimens of each species were selected and mounted in Hoyer’s medium for observation under a microscope, and some specimens were cleared in Nesbitt’s fluid. The remaining samples were stored in 70% ethyl alcohol. The slides were observed under two microscopes: an Olympus BX51 TF (Olympus Group, Tokyo, Japan), with multiple viewing features and phase contrast, and a Zeiss model, Axio Imager. A1, with differential interference contrast (DIC). For measurements, we used a U-DA drawing attachment UIS (Universal Infinity System) and a scale calibrated with a slide by Graticules Ltd., Cambridge, UK (1 mm divided in 100 parts). For SEM (scanning electron microscopy), the specimens were dehydrated using a series of ethyl alcohols followed by critical-point drying in CO_2_. They were then mounted on aluminium SEM stubs and coated in an argon atmosphere with 16 nm gold in a sputter coater from Emitech Ltd., Strovolos, Chipre (model K550). SEM observations were made with an FE-SEM Zeiss model Sigma 300 VP (Zeiss, Oberkochen, Germany).

### 2.4. Nomenclature

Drawing from Vargovitsh, 2009 [16], the terminology for *Pygmarrhopalites* used in the descriptions is as follows: we refer to Fjellberg (1984) [17] for the terminology of the outer maxillary palp and to Fjellberg (1999) for the terminology of the labial palp [18]; we refer to Christiansen and Bellinger (1996) [19] for the terminology of the Ant III sensory organ; we refer to Bretfeld (1999) [20] for the terminology for Abd VI; we refer to Christiansen (1966) [21] and Christiansen and Bellinger (1998) [22] for the terminology of the empodium; and we refer to Vargovitsh (2009, 2017) [16,23] for the terminology of the head, body, and legs. The macrochaetotaxy for *Pseudosinella* follows Gisin and Da Gama (1969) [24], Szeptycki (1979) [25], Soto Adames (2010) [26], and Mateos (2008) [27]. The characters defined by Christiansen for *Pseudosinella*, used in a Delta key by Christiansen in Jordana et al. (2024) [28], were used for identification and description. The equivalence between the notation proposed by Gisin (1965, 1967a, b) [29,30,31] and the AMS system *sensu* Soto-Adames 2010 [26] is given in Baquero et al., 2020 [32]. The material has been deposited at the MZNA (Museum of Zoology at the University of Navarra) (Pamplona, Spain).

The codes of the institutions and sample codes are as follows:
MZNAMuseum of Zoology at the University of Navarra, Pamplona, SpainPBAGPablo Barranco’s collections in Cueva del AguaPBAPPablo Barranco’s collections in Cueva de los ApasPBC3Pablo Barranco’s collections in Cueva C 3PBCAPablo Barranco’s collections in Sima del CamiónPBCOPablo Barranco’s collections in Cueva CovaduraPBGEPablo Barranco’s collections in Complejo GEPPBTEPablo Barranco’s collections in Cueva del Tesoro

## 3. Results

### 3.1. Biocenosis

The extensive sampling effort (83 cave visits) resulted in the collection of 7869 specimens, belonging to the four orders of the class Collembola. These include 5261 Symphypleona from four different families—Arrhopalitidae (5214 specimens), Bourletiellidae (10), Sminthuridae (29), and Sminthurididae (8); 2552 Entomobryomorpha from five families—Entomobryidae (1792), Isotomidae (96), Orchesellidae (84), Paronellidae (575), and Tomoceridae (3); 32 Poduromorpha, all belonging to the family Mesogastruridae; and 29 Neelipleona, all from the family Neelidae. In total, 17 genera were studied, and 28 different species were identified, 18 of which have been named, including the 4 species (three new) described in this paper (Table 1).

### 3.2. Taxonomy

Class Collembola Lubbock, 1873 [33];

Order Entomobryomorpha Börner, 1913 [34], sensu Soto-Adames et al., 2008 [35];

Family Lepidocyrtidae Whalgren, 1906 [36] sensu Zang and Deharveng, 2015 [37];

Subfamily Lepidocyrtinae Whalgren, 1906 [36];

Genus *Pseudosinella* Schäffer, 1897 [38].

#### 3.2.1. *Pseudosinella sexocellata* Jordana and Baquero sp. nov.

http://zoobank.org/CECBC14F-3AC9-40D9-A606-2B489F0A992F, accessed on 12 February 2025.

Figure 2A–G, Figure 3A–C and Figure 4A–C.

#### Type Locality

Cueva Covadura, municipal district of Sorbas, Almería, Spain.

#### Type Material

Holotype. Female, 16.x.2001, slide labelled “PBCO-098”, Ruiz-Portero leg. Paratypes, all Ruiz-Portero leg. unless otherwise stated (sample, in tube with ethyl alcohol, on slide): Complejo GEP, PBGE-064, 1, 1. Cueva Covadura, PBCO-086, 1, 1; PBCO-087, 0, 1; PBCO-088, 0, 1; PBCO-089, 0, 1; PBCO-090, 1, 1; PBCO-091, 0, 1; PBCO-092, 5, 1; PBCO-093, 16, 1; PBCO-094, 3, 1; PBCO-095, 68, 1; PBCO-096, 1, 1; PBCO-097, 0, 1; PBCO-099, 7, 1; PBCO-100, 0, 1; PBCO-101, 0, 1; PBCO-102, 0, 1; PBCO-103, 1, 1; PBCO-104, 1, 1; PBCO-105, 59, 1; PBCO-106, 0, 1; PBCO-107, 9, 1; PBCO-108, 13, 1; PBCO-109, 14, 1; PBCO-110, 1, 1. Cueva Apas (Ruiz-Portero and Fernández leg.), PBAP-009, 6, 2; PBAP-010, 0, 2 (Barranco and Amate leg.); PBAP-011, 29, 2; PBAP-012, 182, 2; PBAP-013, 70, 2; PBCO-118, 26, 1. Cueva del Agua, PBAG-001, 0, 1; PBAG-002, 8, 1; PBAG-003, 1, 1. All deposited at the Museum of Zoology, University of Navarra, Pamplona, Spain (MZNA).

**Figure 4 insects-16-00309-f004:**
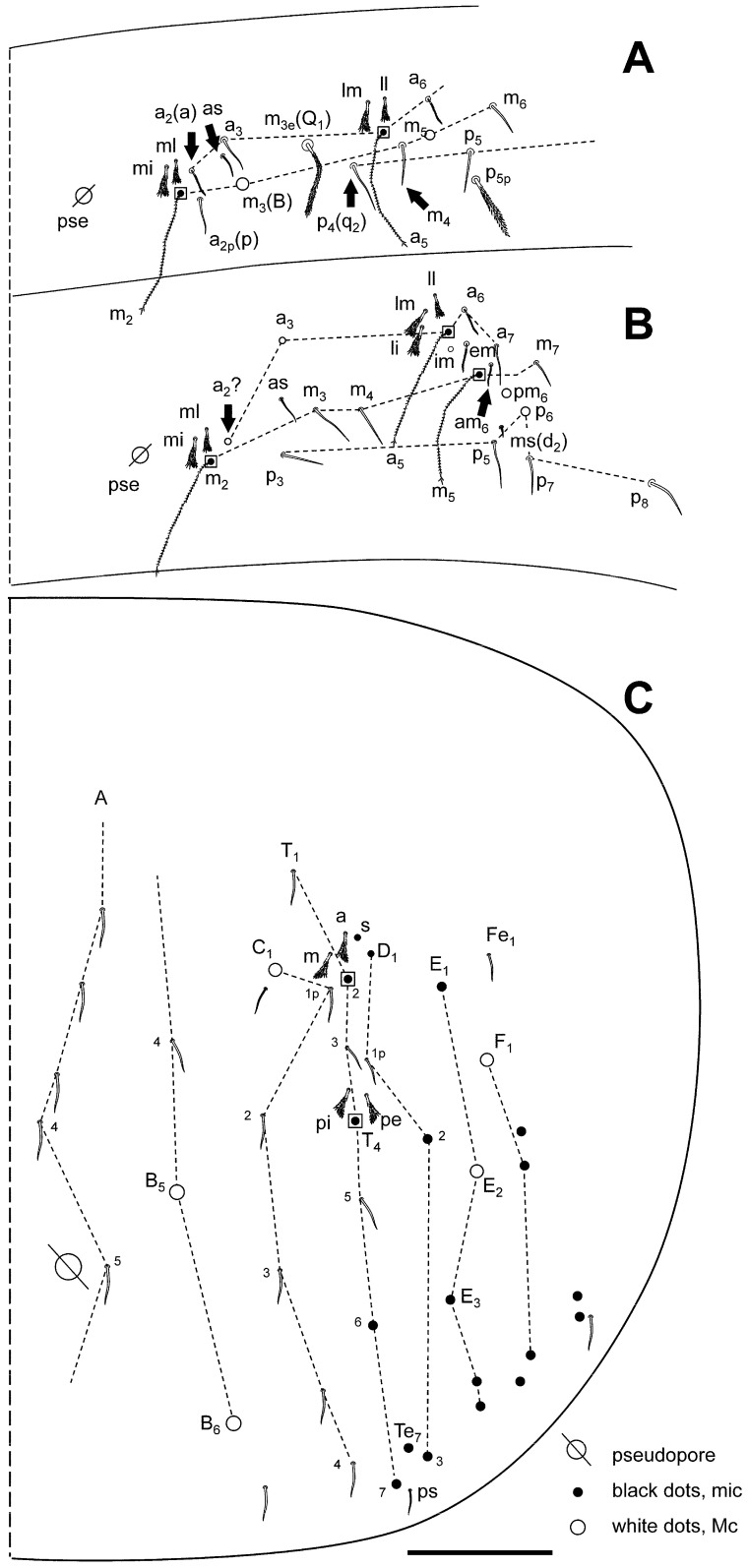
*Pseudosinella sexocellata* sp. nov. macrochaetotaxy. (**A**) Abd II; (**B**) Abd III; (**C**) Abd IV. Scale bar: 0.05 mm.

#### Etymology

The name refers to the total number of eyes on the head.

#### Diagnosis

With 3 + 3 ocelli. Antennae only slightly longer than head. Ant III sense organ with two rod-like and three additional sensilla. Not ringed area of dens 2–3 times the length of mucro. Claws elongated, with four internal teeth; tenent hair capitate. Reduced formula: R_0_R_1_R_2_011/00/0201 + 2/s, paBQ_1_q_2_, M_1_M_2_rEL_1_L_2_.

#### Description

Body length up to 1.34 mm. Colour: white or light yellowish. Only eyes pigmented.

Head with three eyes by side (ABG). Intraocular chaetae p, t, and s present. Only A_0_, A_2_, A_3_, An_1_, An_2_, An_2a-c_, An_3a_, An_3_, S_4_ and Pa_5_ as Mc (Figure 2A). Ratio antenna/cephalic diagonal 1.06–1.40. Antennal segments I/II/III/IV ratios 1/1.5–2.0/1.2–1.6/2.3–3.4 (Figure 2B). Ant IV: apical vesicle absent; in this segment, there are some tiny sensilla on the two distal thirds, smooth and cylindrical, clearly different from the normal chaetae. Ant III sense organ with the common configuration: two rod-like sensilla encased in its pit and more or less one above the other, two guard sensilla one each side, and the last one, the spiny guard sensilla at the other side of the segment (Figure 2B’). Ant I with three small chaetae on its distal part. Antennae without scales. Prelabral chaetae (four) ciliated, labral rows a, m, and p all smooth. Labral papillae with a middle chaeta-like projection. Formula of the labial base M_1_M_2_rEL_1_L_2_ (M_1_ is very wide and with ciliation very evident; M_2_, E, L_1_, and L_2_ apparently smooth, but have the ciliation weaker and appressed; some lack the r chaeta); the remaining chaetae of labium apparently smooth; post-labial area with ciliated chaeta, some with sort fringes, and 1–2 vestigial or “x” chaetae (Figure 2C). Bifurcate maxillary palp with three smooth sublobal chaetae. Labial papilla (l.p.) E with finger-shaped process not reaching the base of apical appendage.

Body. Legs without scales. Only two lateral Mc with big alveoli; other chaetae, probably longer than mic but with smaller alveoli. Trochanteral organ with ca 17–20 chaetae (Figure 2D). Differentiated supra-empodial inner chaeta on hind tibiotarsus well differentiated and acuminate. Dorsal tibiotarsal tenent hair capitate, 0.75 times the length of inner margin of claw. Claw with four internal teeth: the basal paired at different position, approximately 30 and 40% from base, respectively, an unpaired well developed, at 75%, and another unpaired sometimes almost as a notch; lateral teeth and dorsal tooth at the level of paired. Empodium appendage acuminate, serrate externally on its distal half (Figure 2E). Retinaculum with 4 + 4 teeth and one ciliated chaeta. Ventral tube without scales; lateral flap with 7 slightly ciliated chaetae (2 bigger than the rest), and 3 + 3 posterior chaetae. Manubrium and dens with scales only ventrally (anterior); two internal and three external chaetae related to two pseudopores of manubrial plate (Figure 2F); not ringed area of dentes 1.5–2 times the length of mucro; mucro with distal tooth slightly longer than the anteapical; basal spine reaching and surpassing the tip of anteapical tooth (Figure 2G).

Macrochaetotaxy (Figure 3A–C and Figure 4A–C). Th II and Th III without Mc. Abd II: chaetae p, a, and q_2_ as ciliated mic, chaetae B and q_1_ as broad ciliated Mc (q_1_ shorter than B and with a smaller alveoli, and with delicate fringes); mi and ml chaetae over bothriotrichum (m_2_) fan-shaped; lm and ll over bothriotrichum (a_5_) fan-shaped mic. Abd III: mi and ml over bothriotrichum m_2_, and li, lm, and ll over bothriotrichum a_5_ fan-shaped; a_2_ as slightly broadened ciliated mic; ‘as’ in equidistant to a_3_ and p_3_, that are apparently smooth mic, the same as m_3_ and m_4_; im, em, and a_6_ surrounding bothriotrichum a_5_, and am6 next bothriotrichum m_5_ as slightly ciliated pointed mic; a_7_ as mic m_5_ bothriotrichum; pm_6_ and p_6_ as Mc without d_3_ between them; ‘ms’ (d_2_) between p_5_ and p_6_; m_7_, p_7_, and p_8_ as mic. Abd IV: accessory chaeta ‘s’ in the anterior trichobothrial complex present (absent in one asymmetrical specimen). Medial chaeta B_5_ below the level of the trichobothrium T_4_. Pseudopore between B_5_ and B_6_. Reduced formula (from Gisin 1965, 1967a, b) [29,30,31]: R_0_R_1_R_2_011/00/0201 + 2/s, paBQ_1_q_2_, M_1_M_2_rEL_1_L_2_; C_1_, B_5–6_; ratio between C_1_-B_5_/B_5_-B_6_ near 1.00, n = 2; two lateral mac (E_2_ and F_1_); T_5_ as mic; before T_2_ bothriotrichum four ciliated mic (a, m, s, and D_1_).

#### Remarks

There are 31 species described with 3 + 3 eyes, but only 12 have 2 anterior and 1 posterior on each side. Independently (among those with 3 + 3 eyes, regardless of their arrangement), only 12 lack Mc in Th II, and among them, only 4 have chaeta p present on Abd II (*P. gutierrezae* Simón-Benito & Palacios-Vargas, 2008, and *P. torcuatoensis* Simón-Benito & Palacios-Vargas, 2008, both from La Rioja, Spain [39]; *P. ops* Christiansen & Bellinger, 1998, from Virginia, USA [22]; and *P. sexoculata* Schött, 1902, from North USA [40] and *P. sexoculata* from Europe: Sweden, Finland, England, France, Austria, and Spain [41]), but only one of them coincides with the new species in the presence of Q_1_ as Mc (*P. sexoculata* from USA, Mc smooth; *P. sexoculata* from Europe, Mc ciliate). It is distinguished from *P. sexoculata* (both American and European specimens, which probably belong to different species) by the chaetotaxy of the labium: M_2_, E, L_1_, and L_2_ are completely smooth in *P. sexoculata*, whereas they are ciliated in the new species. The differences in many other characters between these species are presented in Table 2.

#### 3.2.2. *Pseudosinella najtae* Jordana and Baquero, 2017, in Jordana et al., 2017 [42]

Figure 5A–G and Figure 6.

Studied material. Spain (all Ruiz-Portero leg. unless otherwise stated): Cueva C-3, PBC3-005, 15, 2; PBC3-007, 28, 1; PBC3-008, 27, 1; PBC3-009, 0, 1; PBC3-010, 27, 2; PBC3-011, 6, 2. PBC3-012, 15, 2. Complejo GEP; PBGE-039, 3, 1; PBGE-040, 5, 2; PBGE-041, 16, 2; PBGE-042, 0, 1; PBGE-043, 1, 2; PBGE-044, 0, 1; PBGE-045, 4, 1; PBGE-046, 5, 1; PBGE-047, 126, 2; PBGE-048, 14, 2; PBGE-049, 0, 2; PBGE-050, 0, 1; PBGE-051, 0, 1; PBGE-052, 0, 3; PBGE-053, 0, 1; PBGE-054, 0, 3; PBGE-055, 0, 1; PBGE-056, 13, 2; PBGE-057, 0, 1; PBGE-058, 0, 1; PBGE-059, 0, 1; PBGE-060, 0, 1; PBGE-061, 0, 1; PBGE-062, 0, 1; PBGE-063, 0, 1; PBGE-065, 22, 2. Cueva Covadura, PBCO-113, 2, 1; PBCO-115, 2, 1; PBCO-116, 3, 1; PBCO-117, 0; PBCO-119, 6, 1; PBCO-120, 22, 1. Cueva Apas (Barranco and Amate leg.), PBAP-007, 11, 2; PBAP-008, 0, 1; PBAP-006, 0, 1. Cueva del Tesoro, PBTE-060, 0, 1; PBTE-061, 5, 1; PBTE-062, 0, 1; PBTE-063, 1, 2; PBTE-064, 1, 1; PBTE-065, 0, 1; PBTE-066, 20, 2; PBTE-067, 11, 2; PBTE-068, 208, 2; PBTE-069, 1, 1; PBTE-070, 0, 1; PBTE-071, 2, 1; PBTE-072, 2, 2; PBTE-073, 18, 2; PBTE-075, 0, 2; PBTE-076, 202, 2; PBTE-077, 4, 2; PBTE-078, 148, 2; PBTE-079, 3, 2; PBTE-080, 3, 1; PBTE-081, 0, 1. Sima del Camión, PBCA-009, 7, 1; PBCA-010, 14, 2; PBCA-011, 2, 1; PBCA-012, 0, 1; PBCA-014, 0, 1.; PBCA-013, 17, 1.

**Figure 5 insects-16-00309-f005:**
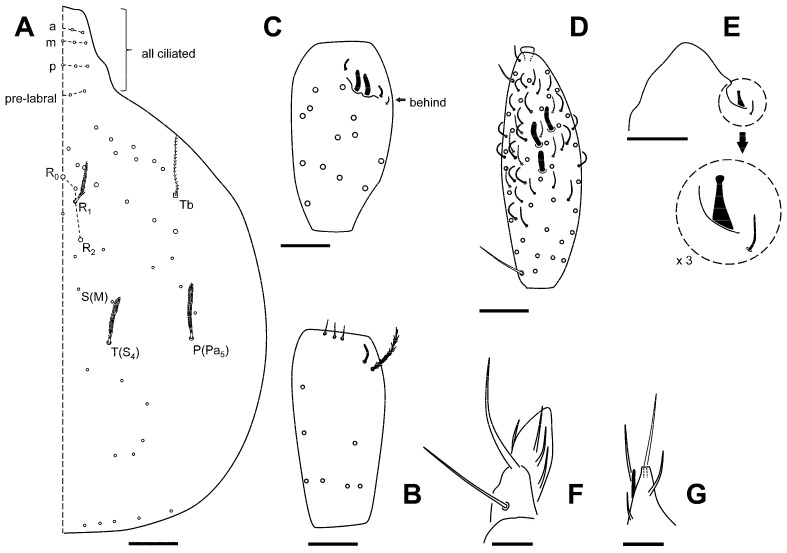
*Pseudosinella najtae*. (**A**) Head, dorsal; (**B**) antennal segment II; (**C**) antennal segment III; (**D**) antennal segment IV, ventral view; (**E**) tip of antennal segment IV, with detail of the special organite; (**F**) maxillary palp; (**G**) labial papilla (**E**). Scale bar: (**A**–**D**), 0.02 mm; (**E**–**G**), 0.01 mm.

**Figure 6 insects-16-00309-f006:**
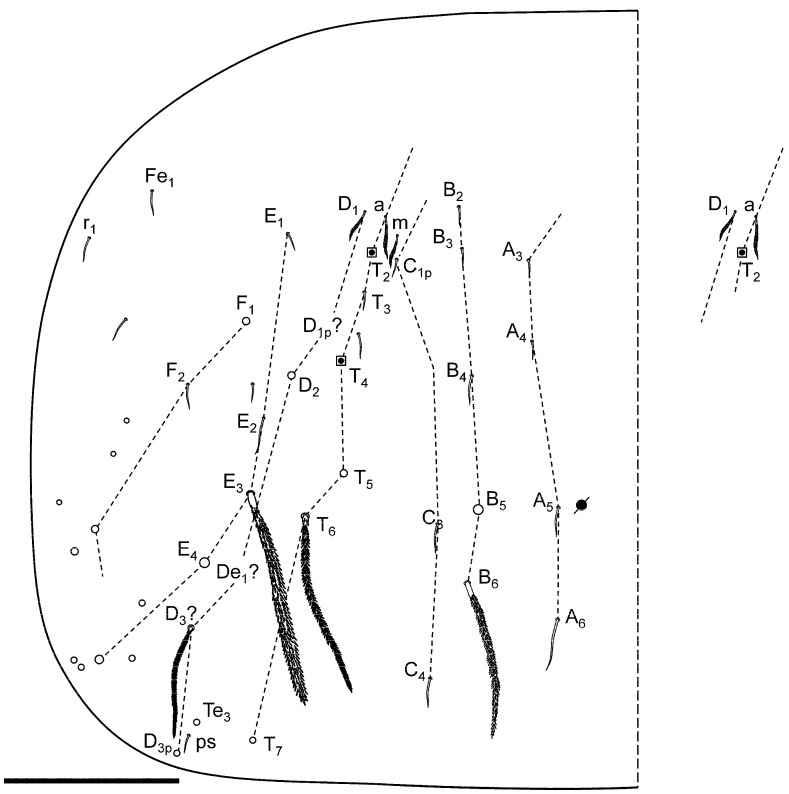
*Pseudosinella najtae*. Abdominal tergite IV macrochaetotaxy. Scale bar: 0.05 mm.

#### Additional Data to Original Description

This species was originally described from Cueva del Saliente and also found in Almería, which is 45 km from GKS. In the original description (Jordana & Baquero, 2017) [42], the maximum size mentioned was 0.82 mm; in the caves in the studied area (GKS), some specimens are bigger, up to 0.99 mm (head and body, excluding antennae). Furthermore, some specimens that could not be identified initially had 2 + 2 eyes, and so the species can be described with the current data as having 0 + 0 or 2 + 2 eyes. There is no correlation between the presence of eyed specimens and the caves in which they appear. The proportion of eyed specimens is low, estimated at 10–20%, and given the large number of specimens studied (1140), it was indeed a stroke of luck to find that they existed (only a small proportion of them, around 10%, were mounted). Some specimens of the Yesos’s caves lack the m chaeta above T_2_ bothriotrichum on Abd IV, sometimes at both sides, and sometimes asymmetrically.

Family Paronellidae Börner, 1906 [43] *sensu* Zhang et al., 2019 [44];

Subfamily Paronellinae Börner, 1906 [43] *sensu* Zhang et al., 2019 [44];

Tribe Paronellini Börner, 1906 [43] *sensu* Zhang et al., 2019 [44];

Genus Troglopedetes Joseph, 1872 [45].

#### 3.2.3. *Troglopedetes machadoi* Delamare-Debouteville, 1946 [46]

Figure 7A–E.

Studied Material. Spain (all Ruiz-Portero leg. unless otherwise stated): Cueva C-3, PBC3-013, 0, 1; PBC3-014, 28, 2; PBC3-015, 34, 2; PBC3-016, 0, 1; PBC3-017, 0, 1; PBC3-018, 7, 2; PBC3-019, 20, 2; PBC3-020, 18, 2; PBC3-021, 30, 2; PBC3-022, 0, 1; PBC3-023, 18, 1; PBC3-024, 0, 1; PBC3-025, 0, 2 (Ruiz-Portero & Fernández leg.). Complejo GEP, PBGE-066, 0, 1; PBGE-067, 22, 2; PBGE-068, 0, 1; PBGE-069, 11, 2; PBGE-070, 1, 1; PBGE-071, 2, 2; PBGE-072, 15, 1; PBGE-073, 0, 2; PBGE-074, 1, 2; PBGE-075, 5, 1; PBGE-076, 34, 2; PBGE-077, 6, 1; PBGE-078, 4, 1; PBGE-079, 37, 2; PBGE-080, 0, 1; PBGE-081, 0, 2; PBGE-082, 0, 1; PBGE-083, 0, 2; PBGE-084, 18, 2; PBGE-085, 2, 2; PBGE-086, 1, 2; PBGE-087, 0, 1; PBGE-088, 5, 2; PBGE-089, 0, 1; PBGE-090, 0, 1; PBGE-091, 0, 1; PBGE-092, 0, 1; PBGE-093, 0, 1; PBGE-094, 3, 1; PBGE-095, 46, 2; PBGE-096, 0, 1; PBGE-097, 1, 1. Cueva Covadura, PBCO-014, 0, 1; PBCO-015, 1, 1; PBCO-016, 6, 1; PBCO-017, 2, 1; PBCO-018, 0, 1; PBCO-019, 13, 1; PBCO-021, 0, 1; PBCO-021, 0, 1; PBCO-023, 2, 1; PBCO-024, 0, 1; PBCO-025, 1, 1; PBCO-026, 4, 1; PBCO-029, 0, 1; PBCO-27, 2, 2; PBCO-28, 9, 1. Cueva de los Apas, PBAP-002, 0, 1 (Barranco y Amate leg.). Cueva del Agua, PBAG-034, 0, 1; PBAG-035, 2, 1; PBAG-036, 0, 1; PBAG-037, 2, 1; PBAG-038, 10, 1; PBAG-039, 3, 1; PBAG-040, 0, 1; PBAG-041, 2, 1; PBAG-042, 1, 1; PBAG-043, 3, 1. Cueva del Tesoro, PBTE-001, 1, 1; PBTE-002, 0, 1; PBTE-003, 4, 1; PBTE-004, 4, 1; PBTE-005, 0, 1; PBTE-006, 0, 1; PBTE-007, 0, 1; PBTE-008, 1, 1; PBTE-009, 18, 1; PBTE-010, 2, 1; PBTE-011, 0, 1. Sima del Camión, PBCA-015, 0, 1; PBCA-016, 0, 1; PBCA-017, 5, 1.

**Figure 7 insects-16-00309-f007:**
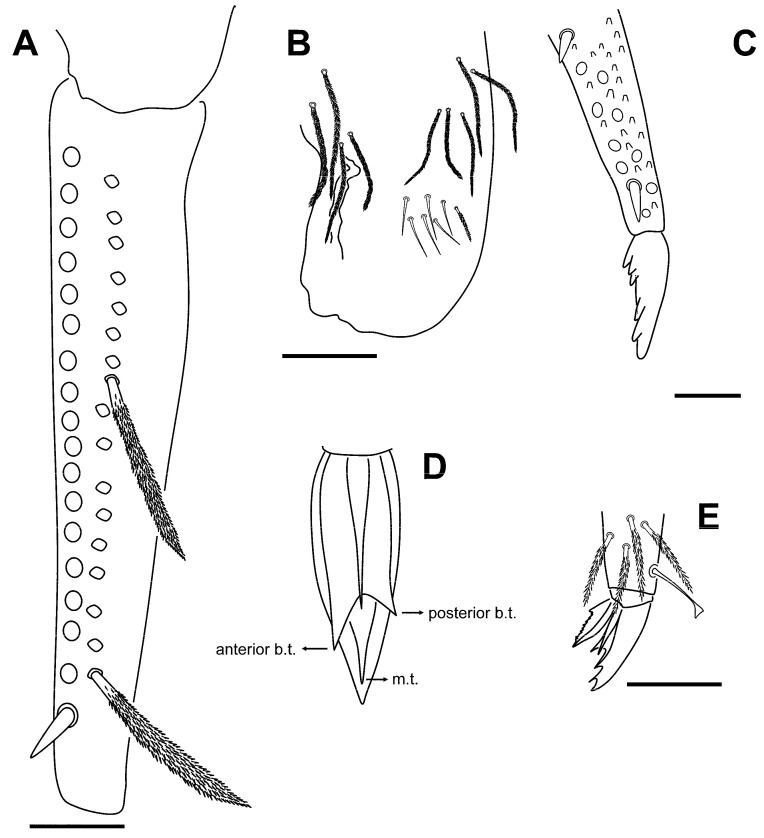
*Troglopedetes machadoi*. (**A**) Dens showing one of the spine-like chaetae, and a special chaeta, each on its single row; (**B**) ventral tube; (**C**) terminal area of dentes and mucro; (**D**) claw of leg 3, ventral view; (**E**) claw and empodium, lateral view. Scale bar: (**A**,**C**,**E**): 0.02 mm; (**B**), 0.05 mm.

#### Remarks

The species was originally described by Delamare-Debouteville [46] from some caves in Portugal (Algar do Pinheiro, Algarve do Cabeço Redondo, and Gruta das Alcobertas). Later, it was recorded from two caves in Tenerife (Canary Islands)—Cueva del Yeso o de las Goteras and Cueva Grande de Chío [47]—and two caves in Almería—Cueva de la Corraliza and Cueva Simarrón II [48]. Regarding posterior references, some of them without official publication (including some of the specimens studied here) have been registered in a data paper [49].

The principal features of the species, shared with the studied specimens, are as follows: antennae shorter than body length; Ant IV without apical bulb; no eyes; labral papillae as rounded projections; lateral process of labial papilla E just exceeding (or slightly) the apex of papilla; basal labial row with chaeta r vestigial, and l_2_ smooth; tenent hairs clavate; unguis with four teeth: I–III basal teeth unequal; medial tooth not surpassing the unguis apex; unguiculus pe lamella serrated (in the original description is drawing smooth); dens with 12–16 inner spines in one row; and mucro in total with 5–6 teeth (2–3 basal and three distal). Only two species have enough similarity to be compared with them: *T. canis* Christiansen, 1957, from Lebanon (with a smaller mucro, 5–6 basal and 2 distal teeth on mucro), and *T. laticlavatus* Stach, 1960, from Afghanistan (with only three teeth on inner unguis, tenent hair pointed, two distal teeth on mucro, and unguiculus not serrated and with one tooth).

Some characters observed in the specimens from the caves of Almería studied are presented in Figure 7: dentes (Figure 7A), ventral tube (Figure 7B), mucro (Figure 7C), and unguis (Figure 7D,E).

Order Symphypleona Börner, 1901 [50] sensu Bretfeld, 1986 [51];

Suborder Appendiciphora Bretfeld, 1986 [51];

Superfamily Katiannoidea Bretfeld, 1994 [52];

Family Arrhopalitidae Stach, 1956 [53];

Genus *Pygmarrhopalites* Vargovitsh, 2009 [16].

#### 3.2.4. *Pygmarrhopalites ruizporteroae* Baquero and Jordana, sp. nov.

http://zoobank.org/B317147C-F6D5-4FA1-A0AF-BD600FC23248, accessed on 12 February 2025.

Figure 8A–E, Figure 9A,B, Figure 10A–C, Figure 11A–D, Figure 12A–F and Figure 13A–C.

#### Type Locality

Cueva del Tesoro, municipal district of Sorbas, Almería, Spain.

#### Type Material

Holotype: female, 28.xii.2001, slide labelled “PBTE-047”, Ruiz Portero leg. Paratypes, all Ruiz Portero leg. unless otherwise stated: Complejo GEP, PBGE-099, 11, 2; PBGE-102, 74, 2; PBGE-104, 0, 1; PBGE-105, 50, 2; PBGE-113, 0, 1; PBGE-114, 11, 2; PBGE-115, 45, 2; PBGE-116, 0, 1; PBGE-117, 62, 2; PBGE-118, 2, 2; PBGE-119, 0, 1; PBGE-120, 52, 2; PBGE-121, 8, 2; PBGE-122, 3, 1; PBGE-123, 10, 2; PBGE-124, 25, 2; PBGE-125, 22, 2; PBGE-126, 153, 2; PBGE-127, 2, 2; PBGE-128, 21, 2; PBGE-129, 5, 2; PBGE-130, 96, 2; PBGE-131, 9, 2; PBGE-132, 11, 2; PBGE-133, 1, 1; PBGE-134, 132, 2; PBGE-135, 195, 2; PBGE-136, 30, 2; PBGE-137, 188, 2; PBGE-138, 43, 2; PBGE-139, 274, 2; PBGE-140, 144, 2; PBGE-141, 35, 2; PBGE-142, 1, 2; PBGE-143, 77, 2; PBGE-144, 0, 2; PBGE-145, 0, 2; PBGE-146, 0, 1; PBGE-147, 0, 1; PBGE-148, 0, 2; PBGE-149, 0, 1; PBGE-150, 0, 1; PBGE-153, 14, 1; PBGE-157, 0, 1; PBGE-158, 27, 1; PBGE-169, 38, 1. Cueva Covadura, PBCO-005, 0, 1; PBCO-011, 1, 2; PBCO-012, 0, 1; PBCO-013, 22, 1; PBCO-032, 0, 1; PBCO-033, 0, 1; PBCO-034, 19, 2; PBCO-035, 0, 1; PBCO-036, 3, 2; PBCO-037, 3, 1; PBCO-038, 3, 1; PBCO-039, 0, 1; PBCO-040, 2, 1; PBCO-041, 13, 2; PBCO-042, 183, 2; PBCO-043, 0, 1; PBCO-044, 13, 2; PBCO-045, 0, 1; PBCO-046, 0, 1; PBCO-047, 0, 1; PBCO-048, 25, 2; PBCO-049, 0, 1; PBCO-050, 0, 1; PBCO-051, 1, 1; PBCO-052, 43, 2; PBCO-053, 27, 2; PBCO-054, 6, 2; PBCO-055, 43, 2; PBCO-056, 7, 2; PBCO-057, 22, 2; PBCO-058, 16, 2; PBCO-059, 7, 2; PBCO-060, 10, 1; PBCO-061, 0, 1; PBCO-062, 140, 2; PBCO-063, 134, 2; PBCO-064, 2, 2; PBCO-065, 6, 2; PBCO-066, 6, 2; PBCO-067, 4, 2; PBCO-068, 0, 1; PBCO-069, 0, 1; PBCO-070, 12, 2; PBCO-071, 1, 1; PBCO-072, 1, 1; PBCO-073, 31, 2; PBCO-074, 3, 2; PBCO-075, 8, 2; PBCO-076, 0, 1; PBCO-077, 67, 2; PBCO-078, 5, 2; PBCO-079, 22, 2; PBCO-080, 19, 2; PBCO-081, 8, 2. Cueva de los Apas, PBAP-014, 0, 1. Cueva del Agua, PBAG-008, 122, 2; PBAG-009, 213, 2; PBAG-010, 15, 2; PBAG-011, 34, 2; PBAG-012, 258, 2; PBAG-013, 0, 1; PBAG-014, 0, 1; PBAG-015, 0, 1; PBAG-016, 2, 1; PBAG-017, 0, 2 (Ruiz-Portero & Fernández leg.); PBAG-018, 9, 2; PBAG-019, 43, 2. Cueva del Tesoro, PBTE-027, 3, 2; PBTE-028, 0, 2; PBTE-029, 35, 1; PBTE-030, 0, 2; PBTE-031, 8, 2; PBTE-032, 2, 2; PBTE-033, 93, 2; PBTE-034, 27, 2; PBTE-035, 5, 2; PBTE-036, 1, 1; PBTE-037, 30, 2; PBTE-038, 0, 2; PBTE-039, 0, 1; PBTE-040, 7, 1; PBTE-042, 0, 1; PBTE-043, 4, 1; PBTE-044, 4, 1; PBTE-045, 23, 1; PBTE-047, 5, 2; PBTE-048, 5, 1; PBTE-049, 43, 1; PBTE-051, 0, 1; PBTE-052, 0, 1; PBTE-057, 2, 1. Sima del Camión, PBCA-001, 0, 1; PBCA-002, 37, 1; PBCA-003, 38, 2; PBCA-004, 1, 1; PBCA-005, 1, 2; PBCA-006, 17, 2; PBCA-007, 16, 2.

**Figure 10 insects-16-00309-f010:**
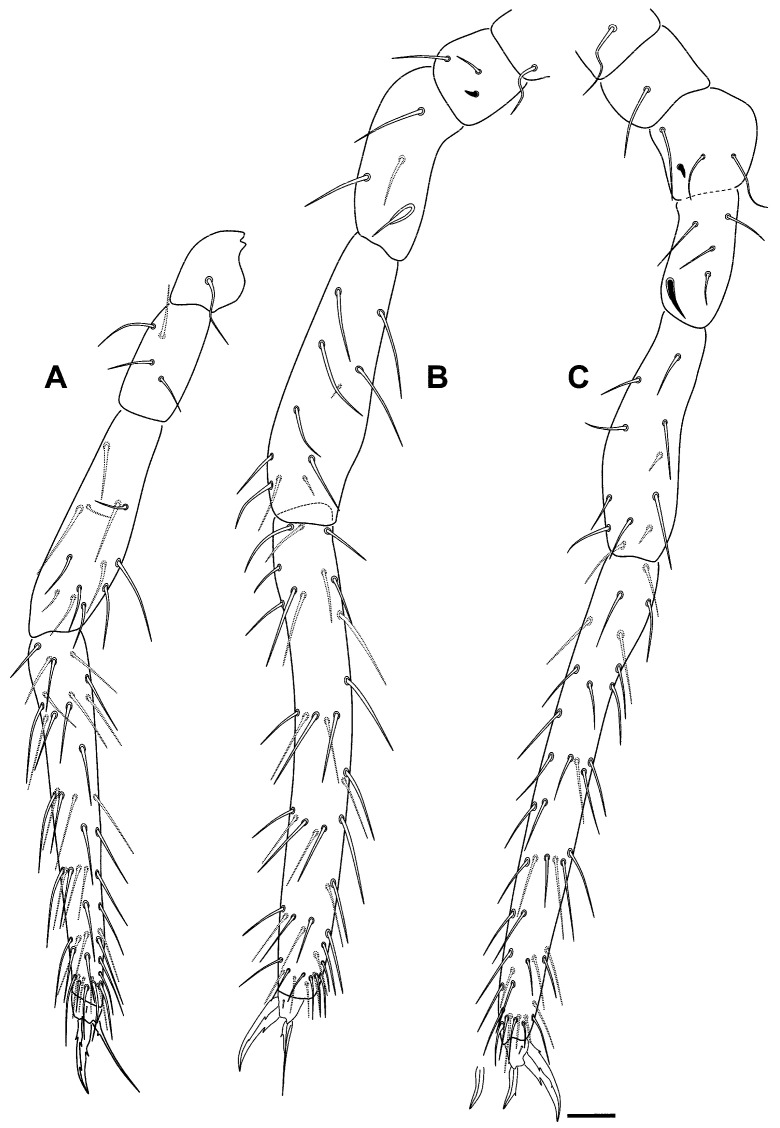
*Pygmarrhopalites ruizporteroae* sp. nov. legs. (**A**) Leg 1; (**B**) leg 2; (**C**) leg 3. Scale bar: 0.025 mm.

#### Etymology

The name refers to Carmen Ruiz Portero, one of the researchers who carried out the survey sampling of the cave fauna at GKS.

#### Diagnosis

Species with 1 + 1 eyes, head without spines, Ant IV with five subsegments, only one thickened spine-shaped chaeta in the dens (external: Ie), anal appendage massive and straight, with few apical denticles, and with very long dorsal chaetae (about six times longer than anterior dorsal chaetae) on the posterior half of the great abdomen.

#### Description

Body length: head, 0.3 mm; body, 0.7 mm (0.6–0.8 mm, n = 5).

Head. Eyes 1 + 1, unpigmented. Clypeal area, row a: 4 + 4 and without axial chaeta; row b: 6 + 6; row c: 5 + 5; row d: 7 + 7; row e: 5 + 5; row f: 7 + 7 (there are one to three additional chaetae between rows d and e). Inter-antennal area, row α: 2 + 2; row β: 1 + 1 and an axial chaeta; rows A and C: 2 + 2 and an axial chaeta; row B: 1 + 1 and an axial chaeta; row D: 2 + 2 chaeta. Lateral chaetae of rows C and D not spine-like (Figure 8A,A’). Labrum: pre-labral/labral chaetotaxy: 6/554 (Figure 8B), all chaetae smooth; a-row central chaeta longer; a-row lateral chaeta shorter and wider than central one. Labium and posterior area in Figure 8C,D; 2 + 2 chaetae near the ventral groove. Maxilla: apical chaeta of the maxillary outer lobe with a short and thin subparallel branch at the base; sublobal plate with three sublobal hairs (Figure 8E).

Antenna (Figure 9A,B): I/II/III/IV, 1/2.25/4.00/11.25; shorter than the body and ratio Ant/head as 1.75 (n = 5); basal subsegment of Ant IV longer than Ant III; mean of the real measurements of Ant IV subsegments 157/48/46/42/112 micrometres (n = 12). Ant I with seven chaetae, a distal one, smaller and another one, skinny. Ant II with 15 chaetae, two interior ones longer than others. Ant III without a conspicuous papilla (only some specimens with the segment slightly broadened), 14 chaetae, the two usual sensory rods, and two shorter thin setae and small blunt curved sensilla; Ant IV with five distinct subsegments, with four evident whorls: one at the end of first and three on the second to fourth subsegments. Apical subsegment with knobbed subapical organite; one of the chaetae on this area is hooked and has a narrowing from terminal half.

Legs (Figure 10A–C and Figure 12A–C): Foreleg precoxae 1, 2, and coxa with 1, 0, 1 chaetae, respectively. Trochanter with three anterior and one posterior chaetae. Femur with 13 chaetae; a_4_ turned perpendicularly to the longitudinal axis of the segment. Tibiotarsus with 47 chaetae: whorl I with nine chaetae, II–V with 8, 8, 7, and 7 chaetae, respectively; region F with three primary FP chaetae (e, ae, and pe) and some secondary chaetae. Pretarsus with one anterior and one posterior chaetae. Foot complex: claw thin, without tunica, with inner tooth, two pairs of lateral teeth, not evident in all specimens (25 and 60% from claw basis) and dorsal tooth; empodium thin, with basal inner tooth, and long apical filament surpassing the tip of the claw. Midleg coxa with two chaetae and an ms. Trochanter with three chaetae and the typical trochanteral organ. Femur with 11 chaetae, p_1_ and p_3_ very small. Tibiotarsus with 45 chaetae: whorl I with 10 chaetae, whorls II–V with 8, 8, 8, and 7 chaetae, respectively; region F with 3 FP chaetae and FSa chaeta. Foot complex: claw wider than foreleg claw, with tunica not evident, inner tooth, two pairs of small lateral teeth (25 and 75% from claw basis), and a dorsal tooth (75%); empodium with corner tooth and a long apical filament surpassing the tip of the claw. Hind leg coxa with three chaetae and an ms. Trochanter with four chaetae and a trochanteral organ. Femur with 11 chaetae, p_1_ and p_3_ as microchaetae. Tibiotarsus with 43: whorl I with 9 chaetae, whorls II–IV with 8, 8, 7, and 7 chaetae, respectively; region F with 3 FP chaetae and FSa chaeta. Foot complex: claw wider than fore- and midleg claw, with tunica not evident, inner tooth, two pairs of small lateral teeth and dorsal tooth; empodium with or without tooth, and without apical filament. Hundreds of specimens were observed, and some lacked the internal tooth of the claw or the empodium, especially in the claw of the hind leg, as reflected in the figure.

**Figure 11 insects-16-00309-f011:**
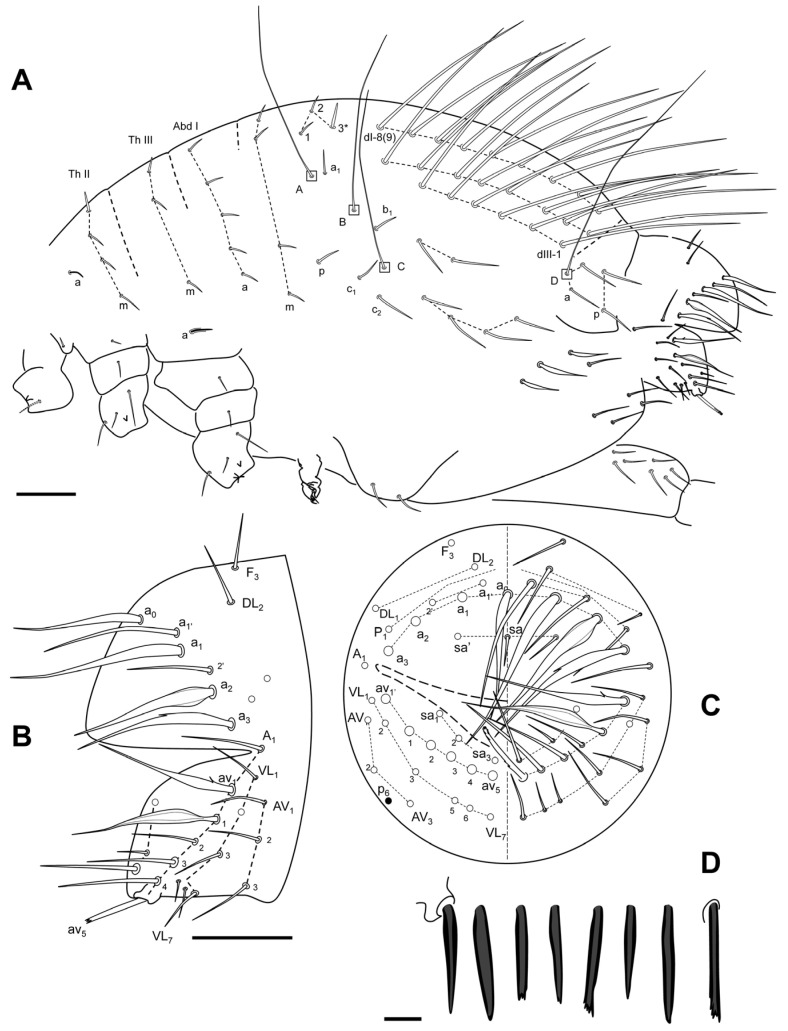
*Pygmarrhopalites ruizporteroae* sp. nov. (**A**) Body of a female, lateral view (the * indicates the chaeta in Figure 12D); (**B**) anal valves; (**C**) schematization of the anal valves, back view; (**D**) some aspects of the anal appendage. Scale bar: (**A**,**B**), 0.05 mm; (**D**), 0.01 mm.

Great abdomen (Figure 11A): Th II with a special sensillum in row a, and four chaetae in row m (m_1_ thickened). Th III with a sensillum in row a and three chaetae in row m. Abd I row a with five chaetae, row m with four, and three p chaetae, above bothriotrichal complex. Bothriotrichal complex: ABC almost linear; bothriotrichum A with one posterior accessory short chaeta; bothriotrichum B with one posterior accessory short chaeta; bothriotrichum C with associated c_1_ and c_2_ chaetae. Posterior lateral complex with 2 + 4 chaetae. Posterior dorsal complex with three rows with 6(7), 8, and 8(9) long chaetae each (mean mucro/chaeta ratio, 0.60; 0.58–0.62; posterior/anterior chaetae 5–6; n = 6). Some of the chaetae under small abdominal are expanded. Sixth abdominal segment (Figure 11B,C): A_0_ not bifurcated, and some of the other circumanal chaetae are broadened, winged, or serrated, some of them lamellate and some bearing single tooth; anal appendage simple, straight with small teeth on its final third (Figure 11D). Tenaculum with two apical chaetae on the corpus, three teeth, and a basal process on each ramus.

**Figure 12 insects-16-00309-f012:**
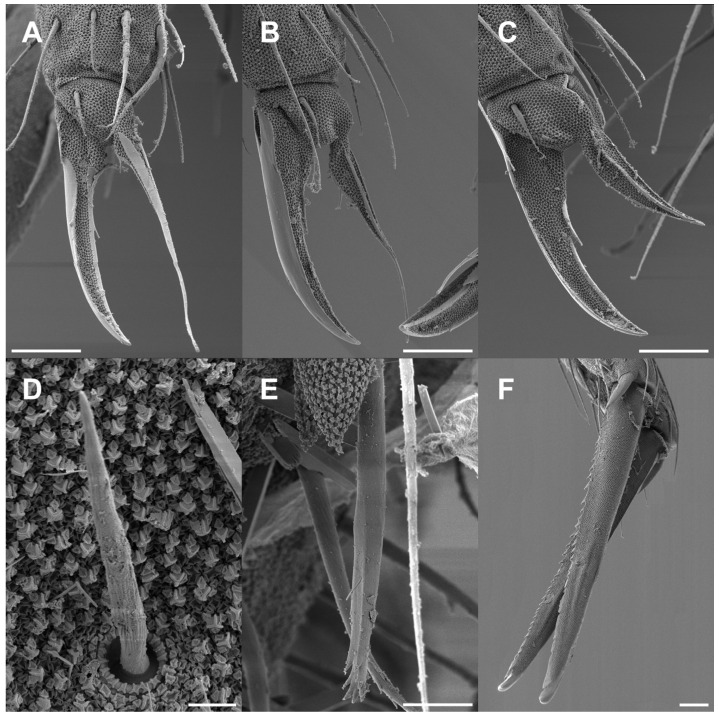
*Pygmarrhopalites ruizporteroae* sp. nov. FE-SEM microphotographs. (**A**) Leg 1; (**B**) leg 2; (**C**) leg 3; (**D**) detail of the chaeta 3 on great abdominal (see Figure 11A); (**E**) anal appendage; (**F**) mucro. Scale bar: (**A**–**C**,**F**), 0.01 mm; (**D**), 0.002 mm; (**E**), 0.005 mm.

Furca (Figure 13A–C): manubrium with 7 + 7 posterior chaetae; dens (23 chaetae or spine-like chaetae): anterior side with 3, 2, 1, 0, 1 chaetae; externally Ie as massive spine, and IIpe as a big spine-like chaeta; internally Ii, IIIp, and IVp moderately spinous, but always with alveoli. Mucro: both lamellae serrated forming a channel between both; tip of mucro pointed (or narrowed). Dens about 1.5–1.7× as long as mucro.

#### Remarks

There are four previously described species that share the presence of only one eye, the absence of spine-like chaetae on the posterior head, the absence of a papilla on Ant III, and a similar shape of the anal appendage (gutter-like with an apex with 3–4 teeth): *P. cantavetulae* Jordana, Fadrique and Baquero, 2012 [54]; *P. crepidinis* Baquero and Jordana 2017 [42]; *P. pygmaeus* (Wankel, 1860) [55] *sensu* Bretfeld 1999 [20]; and *P. maestrazgoensis* Jordana, Fadrique and Baquero, 2012 [54]. The first three have different numbers and positions of spines/spine-like on dens: 203 for external/anterior/internal (the new species has 100). *P. maestrazgoensis* has the dorsal chaetae of great abdominal short, and the chaeta Ie on distal dens provided a long filament.

**Figure 13 insects-16-00309-f013:**
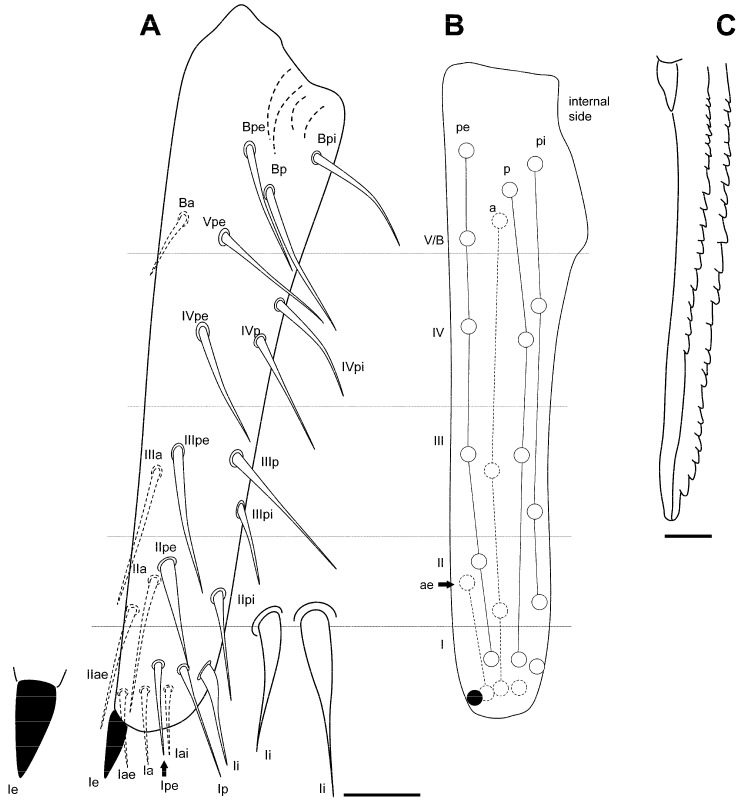
*Pygmarrhopalites ruizporteroae* sp. nov. (**A**) Dens showing the real size of the chaetae; (**B**) dens showing the chaetal rows; (**C**) mucro. Scale bar: (**A**), 0.02 mm; (**C**), 0.01 mm.

#### 3.2.5. *Pygmarrhopalites torresi* Baquero and Jordana, sp. nov.

http://zoobank.org/A7BA9A5D-A76F-4FE9-B54B-6F40C53AD445, accessed on 12 February 2025.

Figure 14A–C, Figure 15A–C, Figure 16 and Figure 17A–C.

#### Type Locality

Cueva Covadura, municipal district of Sorbas, Almería, Spain.

#### Type Material

Holotype: female, 30.iii.2000, slide labelled “PBTE-052”, Ruiz Portero leg. Paratypes, all Ruiz Portero leg. unless otherwise stated: Cueva Covadura, PBCO-002, 11, 2; PBCO-003, 17, 1; PBCO-006, 42, 1; PBCO-007, 0, 1; PBCO-008, 4, 2; PBCO-009, 0, 1; Cueva del Tesoro, PBTE-050, 0, 1; PBTE-053, 11, 2; PBTE-054, 10, 2; PBTE-055, 3, 2; PBTE-057, 0, 1; PBTE-058, 0, 1; Sima del Camión, PBCA-008, 0, 1. Many of the specimens, having been collected in traps that were left there for a long time (the specimens remain on the surface of the liquid and spoil), were not in good condition, and many lacked antennae and legs; therefore, the description was based on the observation of many different specimens.

#### Etymology

The name refers to Angel Torres Palenzuela, speleologist and co-founder of the Espeleo Club Almeria, who has been prospecting the GKS for more than 45 years.

#### Diagnosis

Species with only one eye, head without spines, Ant IV with five subsegments, three thickened spine-shaped chaetae in the dens (two external: Ie and IIIpe, and one internal: Ii), anal appendage massive, straight, and almost without denticles, and with long dorsal chaetae of the posterior half of the great abdomen.

**Figure 14 insects-16-00309-f014:**
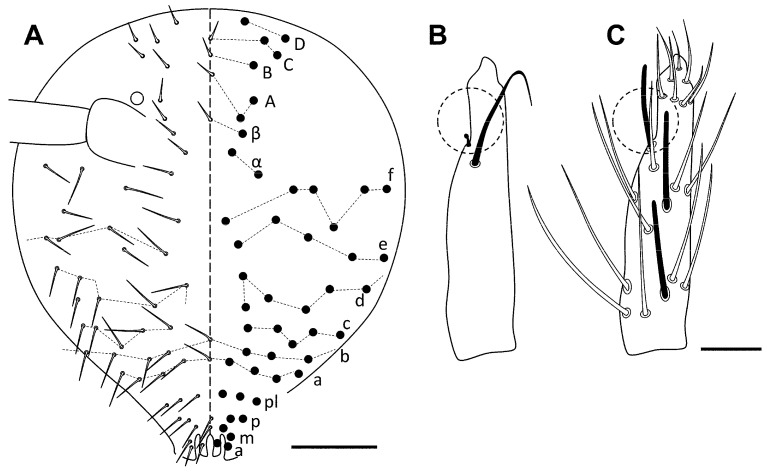
*Pygmarrhopalites torresi* sp. nov. (**A**) Head, ventral; (**B**–**C**) terminal antennal segment IV, views dorsal and ventral. Scale bar: (**A**), 0.05 mm; (**B**,**C**), 0.02 mm.

#### Description

Body length (holotype): head, 0.35 mm; body, 0.7 mm.

Head (Figure 14A). Eyes 1 + 1, unpigmented. Clypeal area, row a: 4 + 4 and an axial chaeta; row b: 3 + 3 and an axial chaeta; row c: 5 + 5; row d: 6 + 6; row e: 5 + 5; row f: 6 + 6, and there is an additional chaeta between rows e and f. Inter-antennal area, row α: 2 + 2; row β: 1 + 1 and an axial chaeta; rows A and C: 2 + 2 and an axial chaeta; row B: 1 + 1 and an axial chaeta; row D: 2 + 2 chaeta. Lateral chaetae of rows C and D not spine-like. Labrum: pre-labral/labral chaetotaxy: 6/554; all chaetae smooth; 2 + 2 chaetae near the ventral groove. Maxilla: apical chaeta of the maxillary outer lobe with a short and thin subparallel branch at the base; sublobal plate with three sublobal hairs.

Antenna (Figure 14B,C): I/II/III/IV, 0.040/0.08/0.125/0.150-0.045-0.42-0.040-0.102; shorter than the body (ratio 0.6) and ratio Ant/head as 1.78; Ant IV with five subsegments; basal subsegment of Ant IV longer than Ant III. Ant I with seven chaetae, distal one smaller and another one skinny. Ant II with 15 chaetae. Ant III without papilla, 14 chaetae (three on the distal area, next to the sensory organ), the two usual sensory rods, and two shorter thin setae and small blunt curved sensilla. Apical subsegment with knobbed subapical organite; one of the chaetae on this area has a narrowing since terminal half.

**Figure 15 insects-16-00309-f015:**
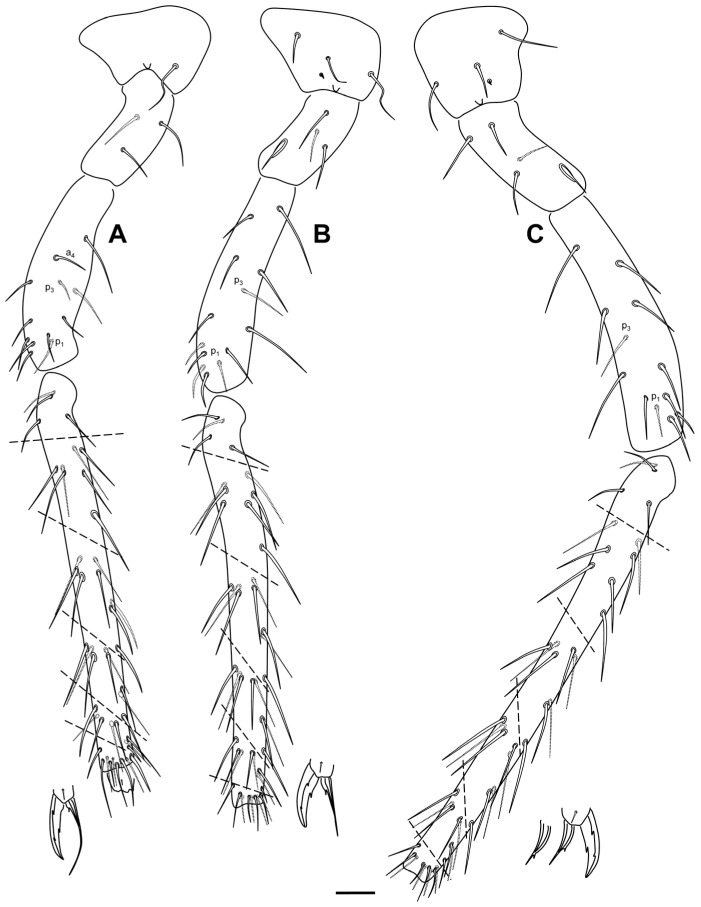
*Pygmarrhopalites torresi* sp. nov. legs. (**A**) Leg 1; (**B**) leg 2; (**C**) leg 3; all with the claw and the empodium next to the distal part of the tibiotarsus. Scale bar: 0.025 mm.

Legs (Figure 15A–C): Foreleg precoxae 1, 2, and coxa with 1, 0, 1 chaetae, respectively. Trochanter with three anterior and one posterior chaetae. Femur with 11 chaetae, a_4_ turned perpendicularly to the longitudinal axis of the segment. Tibiotarsus with 44 chaetae: whorl I with 9 chaetae, II–V with 8, 8, 8, and 7 chaetae, respectively; region F with three primary FP (e, ae, pe) and FSa chaetae. Pretarsus with one anterior and one posterior chaetae. Foot complex: claw narrower than in the middle and hind legs, without tunica, with inner tooth, two pairs of lateral teeth, not evident in all specimens (25 and 60% from claw basis) and dorsal tooth; empodium thin, with basal inner tooth, and long apical filament surpassing the tip of the claw. Midleg trochanter with three chaetae and the typical trochanteral organ. Femur with 13 chaetae, p_1_ and p_3_ similar in length to the other chaetae, but thinner. Tibiotarsus with 44 chaetae: whorl I with 9 chaetae, whorls II–V with 8, 8, 8, and 7 chaetae, respectively; region F with 3 FP chaetae and FSa chaeta. Foot complex: claw wider than foreleg claw, with tunica not evident, inner tooth, two pairs of small lateral teeth (25 and 75% from claw basis), and a dorsal tooth; empodium with corner tooth and a long apical filament surpassing the tip of the claw. Hind leg trochanter with four chaetae and a trochanteral organ. Femur with 12 chaetae, p_1_ and p_3_ not as microchaetae. Tibiotarsus with 44: whorl I with 9 chaetae, whorls II–V with 8, 8, 8, and 7 chaetae, respectively; region F with 3 FP chaetae and FSa chaeta. Foot complex: claw with tunica not evident, inner tooth, two pairs of small lateral teeth and dorsal tooth; empodium with/without tooth (sometimes more distal), and with a short apical filament but not surpassing the tip of the claw.

**Figure 16 insects-16-00309-f016:**
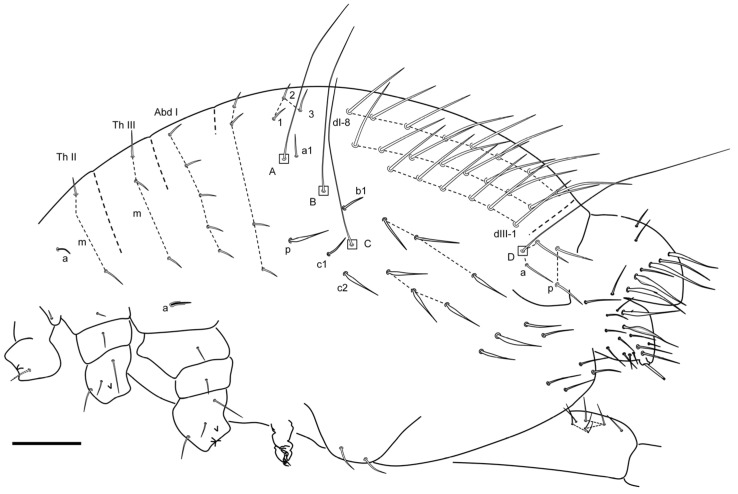
*Pygmarrhopalites torresi* sp. nov. body, lateral (the chaetotaxy of the dorsal valve is not drawn completely). Scale bar: 0.05 mm.

Great abdomen (Figure 16): Th II with a sensillum in row a, and two chaetae in row m. Th III with a sensillum in row a and three chaetae in row m. Abd I row a with five chaetae, row m with four, and three p chaetae, above bothriotrichal complex. Bothriotrichal complex: ABC almost linear; bothriotrichum A with one posterior accessory short chaeta; bothriotrichum B with one posterior accessory short chaeta; bothriotrichum C with associated c_1_ and c_2_ chaeta. Posterior lateral complex with 3 + 3 some expanded chaetae. Posterior dorsal complex with three rows with 6, 8, and 8 long chaetae each (mean ratio mucro/chaeta 0.36; 0.28–0.41, n = 16). Some of the chaetae under small abdominal are expanded. Sixth abdominal segment: A_0_ not bifurcate, and some of the other circumanal chaetae broadened, winged, or serrated; anal appendage simple, straight with small teeth on its final third. Tenaculum with two apical chaetae on the corpus, three teeth, and a basal process on each ramus.

Furca (Figure 17A,C): manubrium with 7 + 7 posterior chaetae; dens (23 chaetae or spine-like chaetae): anterior side with 3, 2, 1, 0, 1 chaetae; Ie, IIIpe (external) and Ii (internal) as massive spines; IIpi and IIIpi as big spine-like chaetae; IIpe moderately spinous. Mucro: both lamellae serrated forming a channel at the end; a little beyond the middle, it undergoes a narrowing, and the lateral denticles become softer, even disappearing, in that area. Dens about 1.3–1.5× as long as mucro.

**Figure 17 insects-16-00309-f017:**
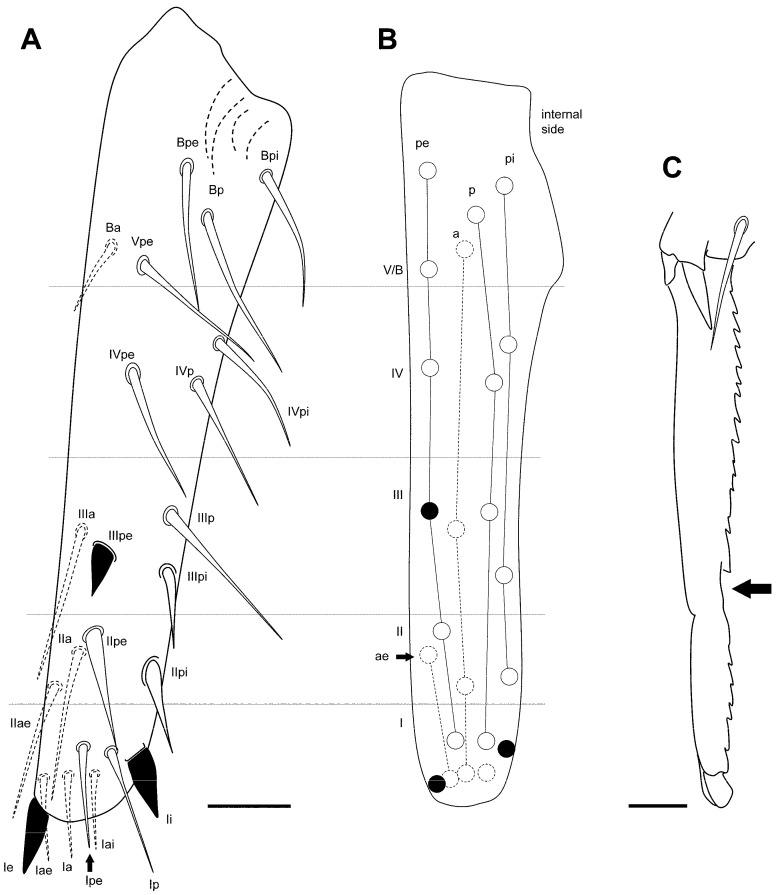
*Pygmarrhopalites torresi* sp. nov. (**A**) Dentes showing the real size of the chaetae; (**B**) dentes showing the chaetal rows; (**C**) mucro. Scale bar: (**A**), 0.02 mm; (**C**), 0.01 mm.

#### Remarks

It shares many of the characters of the species described above, including the characteristic shape of the anal appendage, and for this reason, descriptions of the common characters have been omitted. There are four previously described species that share the presence of only one eye, the absence of spine-like chaetae on the posterior head, the absence of papilla on Ant III, and a similar shape of the anal appendage (gutter-like with an apex with 3–4 teeth): *P. cantavetulae*, *P. crepidinis*, *P. pygmaeus*, and *P. maestrazgoensis*. The first three have different numbers and positions of spines/spine-like on dens: 203 for external/anterior/internal (the new species has 201). *P. maestrazgoensis* has the dorsal chaetae of great abdominal shorter than the new species, and the chaeta Ie on distal dens provided a long filament.

## 4. Discussion

The caves of Karst en Yeso de Sorbas have two very different groups of environments that determine the fauna present and species affinity scarcity. On the one hand, the caves have a more or less constant water regime and are colder and, in turn, there is a subgroup that associates the two caves with a permanent water course, Cueva del Agua and Cueva de los Apas, and another subgroup with irregular water flow, made up of Cueva del Tesoro and Cueva del Yeso (3.5 km away from the group of caves studied but with similar geological characteristics). The other group is made up of caves with a high degree of humidity but without a permanent water regime, which in turn are separated into two subgroups due to their proximity to each other and their location in the same sector—on the one hand, Covadura, Sima los Pinos, and Sima del Camión, and on the other, the GEP Complex with Cueva C-3 [56]. Most of the collembolan species reported and two new species described, *Pseudosinella sexocellata* sp. nov. and *Pygmarrhopalites ruizportaroae* sp. nov., do not seem to show preferences according to any of the different ecological types of the studied caves, but *Pygmarrhopalites torresi* sp. nov. shows a preference for those caves without constant water regimes.

The catch yield is very good, at around 100 specimens/day, and there are cavities that clearly provide more specimens—GEP Complex, C-3, and Cueva del Agua—but not all have a similar richness. C-3 provides a good yield, but only four species are present in it, while in others (GEP Complex, Cueva del Tesoro, or Cueva del Agua) up to 15 and 12 species have been captured out of a total of 28.

The presence of species in caves (which is another way of crossing cave species data) indicates that no species has been found in all the caves and that there are few species present in almost all—*Troglopedetes machadoi*, *Pseudosinella najtae*, *Pseudosinella ruizporteroae* sp. nov— the unidentified species of *Lepidocyrtus*. Most species are present in one or two caves, and the characteristics of the caves cannot be clearly associated with the presence of one species or another. In other words, and considering the totality of the captures and all the species, the different characteristics of the caves do not show a clear correlation.

## Figures and Tables

**Figure 1 insects-16-00309-f001:**
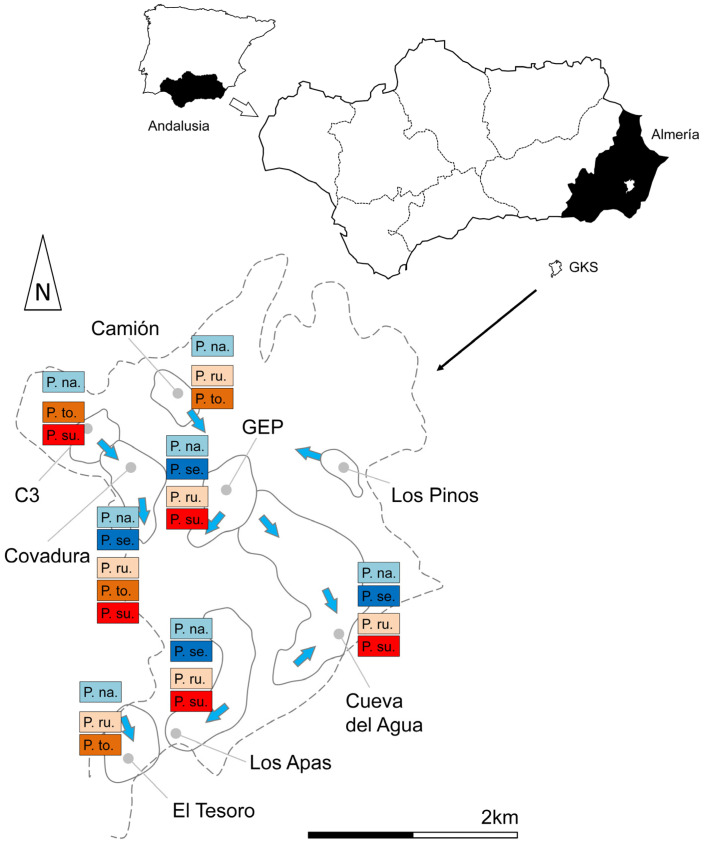
A location map of the caves studied, indicating the known water currents (blue arrows) and therefore the connections between them, as well as the occurrence of species in the caves. P. na., *Pseudosinella najtae*; P. se., *Pseudosinella sexocellata* sp. nov.; P. ru., *Pygmarrhopalites ruizporteroae* sp. nov.; P. to., *Pygmarrhopalites torresi* sp. nov.; P. su., *Pygmarrhopalites subbifidus*; GKS, gypsum karst of Sorbas. The dotted line shows the boundaries of the gypsum outcrop.

**Figure 2 insects-16-00309-f002:**
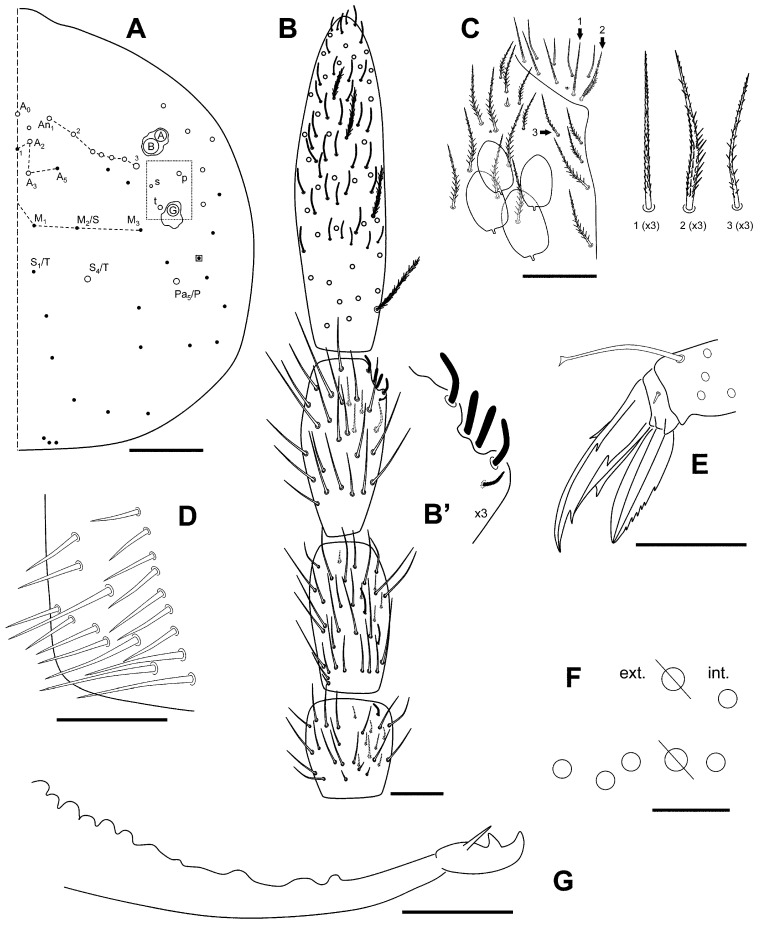
*Pseudosinella sexocellata* sp. nov. (**A**) Head, dorsal; (**B**) antenna (**B’**: detail of the Ant III sensory organ); (**C**) detail of ventral head showing the labial area; (**D**) trochanteral organ; (**E**) distal area of leg 3, showing the claw and empodium; (**F**) chaetae and pseudopores of manubrial plate; (**G**) final area of dentes and mucro. Scale bar: (**A**,**C**), 0.05 mm; (**B**,**D**–**G**), 0.02 mm.

**Figure 3 insects-16-00309-f003:**
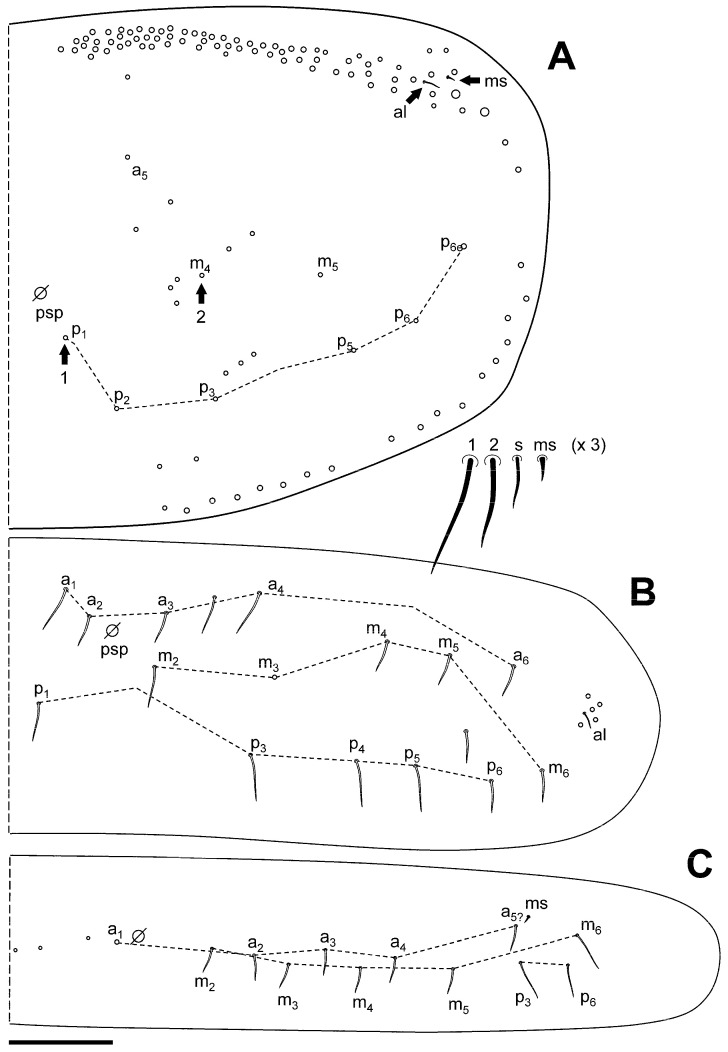
*Pseudosinella sexocellata* sp. nov. macrochaetotaxy. (**A**) Th II, with detail of two chaetae, the sensillum, and the microsensillum; (**B**) Th III; (**C**) Abd I. Scale bar: 0.05 mm.

**Figure 8 insects-16-00309-f008:**
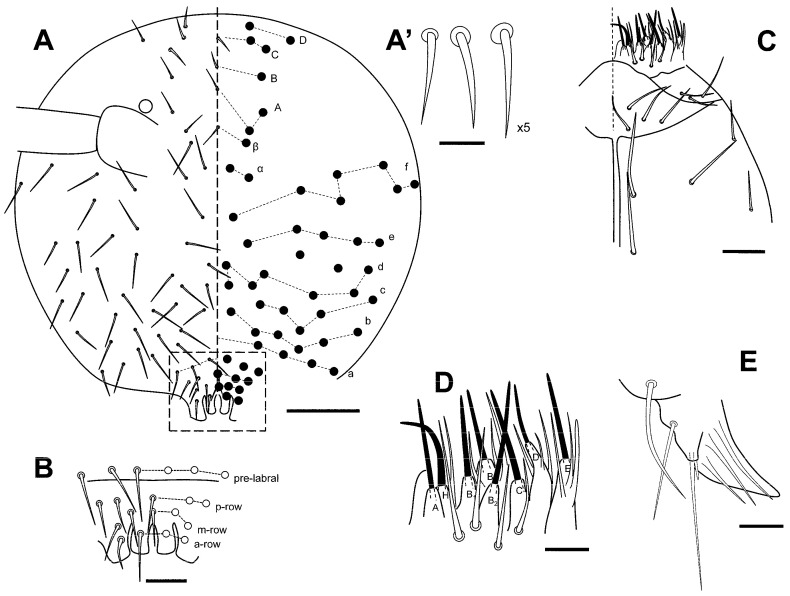
*Pygmarrhopalites ruizporteroae* sp. nov. (**A**) Head, dorsal view, with the detail of the chaetae from vertex (**A’**). The broken line rectangle indicates the area represented in (**B**); (**B**) labral area; (**C**) ventral head, labial area; (**D**) detail of labial papillae; (**E**) maxillary palp. Scale bar: (**A**), 0.05 mm; (**B**,**C**), 0.02 mm; (**A’**,**D**,**E**), 0.01 mm.

**Figure 9 insects-16-00309-f009:**
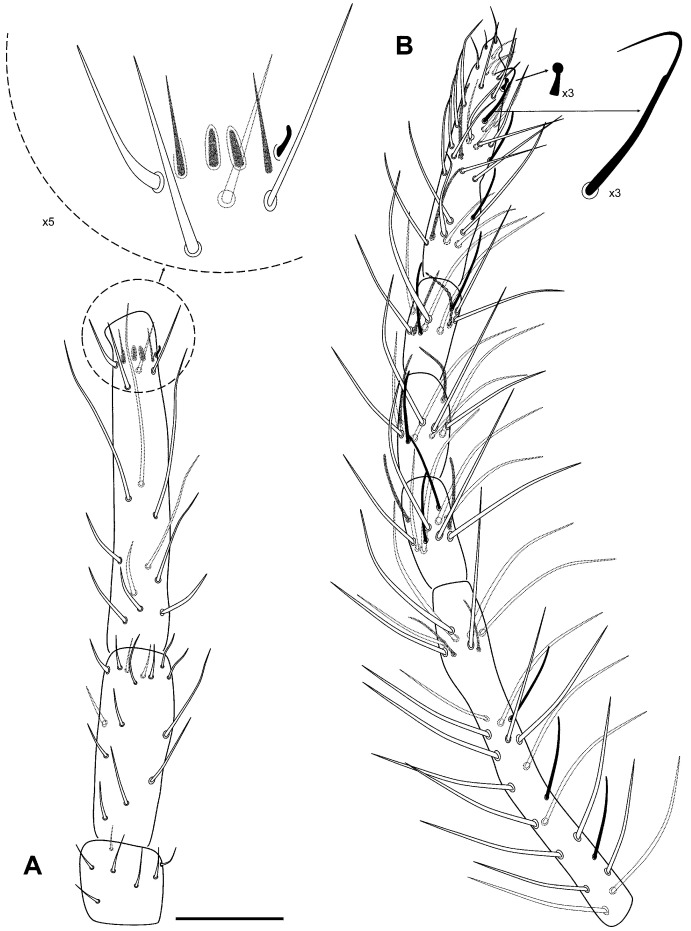
*Pygmarrhopalites ruizporteroae* sp. nov. (**A**) Antennal segments I–III, with detail of the distal area of antennal segment III; (**B**) antennal segment IV. Scale bar: 0.05 mm.

**Table 1 insects-16-00309-t001:** Numbers of specimens of different taxa found throughout all samplings. Some specimens were not considered in the study because they were very small or damaged; therefore, their identification was challenging. The specimens that are presented in genera without species identification could not be identified due to their poor states of conservation. Cave abbreviations: CO, Covadura; C3, Cueva C-3; CA, Sima del Camión; GE, Complejo GEP; SP, Sima los Pinos; TE, Cueva del Tesoro; AP, Cueva de los Apas; AG, Cueva del Agua.

Order and Species	Caves							
**Entomobryomorpha**	CO	C3	CA	GE	SP	TE	AP	AG
*Entomobrya* sp.				25			2	
*Lepidocyrus* sp.	3		4	24	1	12	7	3
*Pseudosinella najtae*	41	129	47	247		661	15	
*P. sexocellata* sp. nov.	254			2			297	12
*Pseudosinella* sp.							2	
*Seira* sp.				1				
*Folsomia candida*						1		
*Folsomia fimetaria*						1		
*Folsomia* sp.								2
*Parisotoma notabilis*	80			3		5		3
*Subisotoma variabilis*				1				
*Heteromurus major*			16	65				
*Heteromurus nitidus*				1				
*Heteromurus* sp.	1							1
*Troglopedetes machadoi*	57	175	8	260		41	1	33
*Tomocerus* sp.								3
**Neelipleona**								
*Neelus murinus*	2						6	20
*Megalothorax* sp.								1
**Poduromorpha**								
*Mesogastrura lybica*								31
*Mesogastrura ojcoviensis*				1				
**Symphypleona**								
*Deuterosminthurus* sp.					1	9		
*Pseudosinella ruizporteroae* sp. nov.	1037		121	1949		332	1	716
*Pseudosinella torresi* sp. nov.	83		1			41		
*P. pygmaeus* cf.						2		
*Pygmarrhopalites* sp.	17	1		71	4			
*P. subbifidus*	4	405		417			10	2
*Gisinurus malatestai*						29		
*Sphaeridia pumilis*				6		2		
**Analysis**								
Total of individuals (per cave)	1579	710	197	3073	6	1136	341	827
Species number (per cave)	11	4	6	15	3	12	9	12
Average specimens/visit	72	118	49	140	6	76	68	92
Number of visits to the cave	22	6	4	22	1	15	5	9

**Table 2 insects-16-00309-t002:** Differentiating characters of *Pseudosinella* species with 3 + 3 eyes that also lack Mc on Th II and have p chaeta (a_2p_) on Abd II.

	An	T	S	M1	M2	R	E	L1	L2	Aa	Ab	Aq1	Aq2	A4	Te	Cl	Ct	Em	Ma	Mp	Ha	Ml	D
*P. sexocellata* sp. nov.	1	2	2	4	4	1	4	4	4	2	4	4	2	2	2	4	2	1	2	2(3)	1	1.34	-
*P. gutierrezae*	1	2	2	4	2 *	0 *	2 *	2 *	2 *	2	4	4	1 *	2	1 *	4	2	3 *	2	2	2 *	1.25	9
*P. ops*	2 *	1 *	U	4	4(2)	3 *	4(2)	4(3)	4	0 *	4	2 *	0 *	1 *	2	3 *	1 *	1	2	6–8 *	1	2.2	11
*P. sexoculata* (USA)	1	2	1 *	4	2 *	0 *	2 *	2 *	2 *	1 *	3 *	3 *	1 *	2	2	3 *	1 *	1	3 *	4–10 *	3	1.7	14
*P. sexoculata* (Europe)	1	2	2	4	2 *	1	2 *	2 *	2 *	1 *	4	4	1 *	2	2	3 *	1 *	3 *	U	U	3 *	1.4	10
*P. torcuatoensis*	1	2	2	4	2 *	1	4	4	4	2	4	1 *	1 *	2	1 *	4	2	3 *	2	2	2 *	1.1	6

**An**, apical organ Ant III: 1, peg- or rod-like; 2, expanded; 3, paddle-shaped. **T**, chaetae S_2_-S_4_ on head: 1, as Mc; 2, as mic. **S**, chaeta Pa_5_ on head: 1, as Mc; 2, as mic. **M1**, ventral labia Chaeta M_1_ shape: 0, absent; 1, smooth microchaeta; 2, smooth macrochaeta; 3, ciliated microchaeta or mesochaeta; 4, ciliated macrochaeta; 5, smooth macrochaeta with supplementary seta; 6, ciliated macrochaeta with a supplementary seta. **M2**, ventral labial chaeta M_2_ shape: 0, absent; 1, smooth microchaeta; 2. smooth macrochaeta; 3, ciliated microchaeta or mesochaeta; 4, ciliated macrochaeta; 5, smooth macrochaeta with supplementary seta; 6, ciliated macrochaeta with a supplementary seta. **R**, ventral labial chaeta r shape: 0, absent; 1, smooth microchaeta; 2, smooth macrochaeta; 3, ciliated microchaeta or mesochaeta; 4, ciliated macrochaeta. **E**, ventral labial chaeta E shape: 0, absent; 1, smooth microchaeta; 2, smooth macrochaeta; 3, ciliated microchaeta; 4, ciliated macrochaeta. **L1**, ventral labial chaeta L_1_ shape: 0, absent; 1, smooth microchaeta; 2, smooth macrochaeta; 3, ciliated microchaeta; 4, ciliated macrochaeta. **L2**, ventral labial chaeta L_2_ shape: 0, absent; 1, smooth microchaeta; 2, smooth macrochaeta; 3, ciliated microchaeta; 4, ciliated macrochaeta. **Aa**, Abd II a (=a_2_) chaeta shape: 0, absent; 1, smooth microchaeta; 2, ciliated microchaeta; 3, smooth macrochaeta; 4, ciliated macrochaeta; 5, fan shape ciliated microchaeta. **Ab**, Abd II b (=m_3_) chaeta shape: 0, absent; 1, smooth microchaeta; 2, ciliated microchaeta; 3, smooth macrochaeta; 4, ciliated macrochaeta. **Aq1**, Abd II q_1_ (=m_3e_) chaeta shape: 0, absent; 1, smooth microchaeta; 2, ciliated microchaeta; 3, smooth macrochaeta; 4, ciliated macrochaeta. **Aq2**, Abd II q_2_ (=p_4_) chaeta shape: 0, absent; 1, smooth microchaeta; 2, ciliated microchaeta; 3, smooth macrochaeta; 4, ciliated macrochaeta. **A4**, Abd IV anterior lateral P (=C_1_/Sm) shape: 1, as mic: 1; 2, as Mc. **Te**, tenent hair shape: 1, acuminate; 2, clavate; 3, truncate. **Cl**, claw (internal) teeth number: 1, 0; 2, only paired; 3, paired + one unpair; 4, paired + two unpaired. **Ct**, claw wing tooth (lateral): 1, absent; 2, present. **Em**, empodium shape: 1, acuminate; 2, truncate; 3, basally swollen. **Ma**, manubrial plate inner chaetae number. **Mp**, manubrial plate, outer chaetae number. **Ha**, habitat: 1, cave; 2, surface; 3, both cave and surface; 4, MSS. **Ml**, maximum length. *, different from the new species. **D**, number of differences with the new species. U, unknown.

## Data Availability

All data are contained within the article. The collected specimens have been deposited in the collection of the Museum of Zoology, Department of Environmental Biology, University of Navarra (MZNA).

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
