# Peer review of "Collembola from the Gypsum Karst of Sorbas (Almería, Spain), with Descriptions of Three New Species"

_insects, 2025, doi:10.3390/insects16030309_

Round 1
Reviewer 1 Report
Comments and Suggestions for Authors
The fauna of Collembola of the Yesos de Sorbas cave complex (Almeria, Spain) is studied based on the long-term collecting. The list of species and their occurrence in the caves are provided, several new species are described. It is a considerable contribution to our knowledge of cave Collembola. The ms can be published with the Insects without major revision. Some small additions, especially for the Discussion part, would improve the ms.
Review to the ms "Collembola from the Gypsum Karst of Sorbas (Almeria, Spain), with descriptions of three new species" by Enrique Baquero, Pablo Barranco, and Rafael Jordana
The fauna of Collembola of the Yesos de Sorbas cave complex (Almeria, Spain) is studied based on the long-term collecting. The list of species and their occurrence in the caves are provided, several new species are described. It is a considerable contribution to our knowledge of cave Collembola. The ms can be published with the Insects without major revision. Some small additions, especially for the Discussion part, would improve the ms.
I did not correct English since I am not a native English speaker.
My small comments/suggestions:
The Title, Simple summary (as a popular version of Abstract) and Abstract are readable and concise.
I am facing † sign at Enrique Baquero. That means that "these authors contributed equally to this work" (see below in the text of the ms). It looks strange since only Baquero is supplied with this sign. Also, as I know, Enrique did not pass away. Let's replace † with + or so. I am wrong? Up to editorial board.
The Materials and Methods part.
The list of caves supplied (1-st paragraph) with coordinates does not fully conform with the subsequent, more detail, list of the same caves (following paragraphs). They should be the same, if I understand this correctly.
The first list is as: Covadura (C-3), Sima del Camion, Complejo, Sima los Pinos ... etc.
The second list is as: Covadura, The Cueva C3, Sima del Camion, Complejo, Sima los Pinos ... etc.
Can I understand that C-3 is a part of Covadura system? Make it more clear pls.
Choose between "1.-" and "2-" notations.
Below we see such list of abbreviations: PBAG - Pablo Barranco Cueva del Agua, PBAP - Pablo Barranco Cueva de los Apas ... etc. I would rephrase this as about: PBAG - "Cueva del Agua (coll. Pablo Barranco)" or "Pablo Barranco's collections in Cueva del Agua" so, because the "Pablo Barranco Cueva del Agua" does not sound fully correct.
Figure 1. I did not found the reference to the figure ("Fig. 1") in the text. This reference should be both in the Materials and Methods part and the Results part (see below). Looks readable apart from abbreviations of species. I would abbreviate them in more common form like: Pseudosinella sexocellata as P.s. (or Ps.s.), not as PS6, etc. I guess, such abbreviations were used in draft variant. The authors can keep draft abbreviations in the Abbreviation part if wanted. Also, PS0 (P.n. would be better) is omitted for Pseudosinella najtae in abbreviations in the legend to the Figure - pls, insert it. I have found the abbreviation GKS in the figure (very small, near the arrow) - it is not explained in the legend. The legend to Figure 1 is "Location map of the caves studied, indicating the known water currents, and therefore the 150 connections between them". The figure shows, however, also the occurence of species in the caves. So, first let's change the title, the second the figure needs to be referred in the Discussion and the Results part, and the third I would shift the figure closer to the Results part since it is important for the results (up to the editorial board, of course).
"All three caves are visited by tourists and cavers alike, although the Cueva de los Apas receives comparatively fewer visitors." This deserves the separate paragraph or is to be integrated to the detail list of the caves given above. Not within the description of traps.
"The terminology for Pygmarrhopalites Vargovitsh, 2009 [16] used in descriptions follows Fjellberg (1984) for the outer maxillary palp; ..." is in the ms.
"The terminology for Pygmarrhopalites by Vargovitsh, 2009 [16] is used in descriptions, we follow Fjellberg (1984) for the outer maxillary palp; ..." - in a such way, maybe?
The Results part
I have no any essential comments to this and the main part of the ms. I think, the descriptions and associated figures are perfect.
"Table 1. Taxa found throughout all samplings."
"Table 1. Number of specimens of different taxa found throughout all samplings." - in a such way, maybe?
"Table 2. Characters that separate ... "
"Table 2. Differentiating characters of ... " - in a such way, maybe?
Pseudosinella najtae was described from Saliente Cave, also in Almeria. I would mention how far the type locality is from the caves where the species is recorded now.
The Discussion part
- The main portion of the text of the Discussion part is devoted to the description of difference between the caves: "The caves of Karts en Yeso de Sorbas have two very different groups of environments that determine the fauna present and whose species affinity is scarce ..." etc.
As I can understand all the caves can be divided into two groups, and then each group can be divided into two subgroups. I would move this to the Materials and Methods part where these characteristics of caves are "hidden" within their descriptions. This division should be formally exposed, as about:
Group 1 ...
- subgroup ... (name of the cave(s))
- subgroup ... (name of the cave(s))
Group 2 ...
- subgroup ... (name of the cave(s))
- subgroup ... (name of the cave(s))
- The discussion is about the distribution of cave species in the studied cave. The distribution should be based on the results. I have expected to see the results in the Table 1 that would be consistent. Table 1, however, does not show any records of species by the caves (is it possible to add?). So, we have the only phrase for the Discussion part "Most collembolan species reported and two new species described, Pseudosinella sexocellata sp. nov. and Pygmarrhopalites ruizportaroae sp. nov. seem not to have preference by any of the different ecological types of caves in the karts (!!), but Pygmarrhopalites torresi sp. nov. shows preference by those caves without constant water regime.". Can author be not so laconic? A little information about other species would be appropriate. So, the discussion is too short, IMHO. Could you expand it a little pls.
See also the "karts" in two places. Is it OK? Karst?
Comments on the Quality of English LanguageConsidering the evaluation of Quality of English Language. I did not correct English since I am not a native English speaker. It is required to choose, however. I have chosen the first option but my choice does not make much sense.
Author Response
Comment 1. The fauna of Collembola of the Yesos de Sorbas cave complex (Almeria, Spain) is studied based on the long-term collecting. The list of species and their occurrence in the caves are provided, several new species are described. It is a considerable contribution to our knowledge of cave Collembola. The ms can be published with the Insects without major revision. Some small additions, especially for the Discussion part, would improve the ms.
Answer 1. Thanks.
Comment 2. Review to the ms "Collembola from the Gypsum Karst of Sorbas (Almeria, Spain), with descriptions of three new species" by Enrique Baquero, Pablo Barranco, and Rafael Jordana
The fauna of Collembola of the Yesos de Sorbas cave complex (Almeria, Spain) is studied based on the long-term collecting. The list of species and their occurrence in the caves are provided, several new species are described. It is a considerable contribution to our knowledge of cave Collembola. The ms can be published with the Insects without major revision. Some small additions, especially for the Discussion part, would improve the ms.
I did not correct English since I am not a native English speaker.
My small comments/suggestions:
The Title, Simple summary (as a popular version of Abstract) and Abstract are readable and concise.
I am facing † sign at Enrique Baquero. That means that "these authors contributed equally to this work" (see below in the text of the ms). It looks strange since only Baquero is supplied with this sign. Also, as I know, Enrique did not pass away. Let's replace † with + or so. I am wrong? Up to editorial board.
Answer 2. Thanks, done (by Enrique, je). The symbol has been supplied by other (‡).
Comment 3. The Materials and Methods part.
The list of caves supplied (1-st paragraph) with coordinates does not fully conform with the subsequent, more detail, list of the same caves (following paragraphs). They should be the same, if I understand this correctly.
The first list is as: Covadura (C-3), Sima del Camion, Complejo, Sima los Pinos ... etc.
The second list is as: Covadura, The Cueva C3, Sima del Camion, Complejo, Sima los Pinos ... etc.
Can I understand that C-3 is a part of Covadura system? Make it more clear pls.
Choose between "1.-" and "2-" notations.
Answer 3. Done, rewritten adding the coordinates to the paragraphs; 1.- has been chosen.
Comment 4. Below we see such list of abbreviations: PBAG - Pablo Barranco Cueva del Agua, PBAP - Pablo Barranco Cueva de los Apas ... etc. I would rephrase this as about: PBAG - "Cueva del Agua (coll. Pablo Barranco)" or "Pablo Barranco's collections in Cueva del Agua" so, because the "Pablo Barranco Cueva del Agua" does not sound fully correct.
Answer 4. Really, done.
Comment 5. Figure 1. I did not found the reference to the figure ("Fig. 1") in the text. This reference should be both in the Materials and Methods part and the Results part (see below). Looks readable apart from abbreviations of species. I would abbreviate them in more common form like: Pseudosinella sexocellata as P.s. (or Ps.s.), not as PS6, etc. I guess, such abbreviations were used in draft variant. The authors can keep draft abbreviations in the Abbreviation part if wanted. Also, PS0 (P.n. would be better) is omitted for Pseudosinella najtae in abbreviations in the legend to the Figure - pls, insert it. I have found the abbreviation GKS in the figure (very small, near the arrow) - it is not explained in the legend. The legend to Figure 1 is "Location map of the caves studied, indicating the known water currents, and therefore the connections between them". The figure shows, however, also the occurence of species in the caves. So, first let's change the title, the second the figure needs to be referred in the Discussion and the Results part, and the third I would shift the figure closer to the Results part since it is important for the results (up to the editorial board, of course).
Answer 5. Some suggestions has been considered. Not the relocation of the figure.
Comment 6. "All three caves are visited by tourists and cavers alike, although the Cueva de los Apas receives comparatively fewer visitors." This deserves the separate paragraph or is to be integrated to the detail list of the caves given above. Not within the description of traps.
Answer 6. Separate paragraph, done.
Comment 7. "The terminology for Pygmarrhopalites Vargovitsh, 2009 [16] used in descriptions follows Fjellberg (1984) for the outer maxillary palp; ..." is in the ms.
"The terminology for Pygmarrhopalites by Vargovitsh, 2009 [16] is used in descriptions, we follow Fjellberg (1984) for the outer maxillary palp; ..." - in a such way, maybe?
Answer 7. No, but now we add “:” for a better compression. Vargovitsh, 2009 [16] is cited as author for the genus name, only.
Comment 8. The Results part
I have no any essential comments to this and the main part of the ms. I think, the descriptions and associated figures are perfect.
Answer 8. Thanks.
Comment 9. "Table 1. Taxa found throughout all samplings."
"Table 1. Number of specimens of different taxa found throughout all samplings." - in a such way, maybe?
Answer 9. Done, thanks.
Comment 10. "Table 2. Characters that separate ... "
"Table 2. Differentiating characters of ... " - in a such way, maybe?
Answer 10. Done.
Comment 11. Pseudosinella najtae was described from Saliente Cave, also in Almeria. I would mention how far the type locality is from the caves where the species is recorded now.
Answer 11. Done, 45 km.
Comment 12. The Discussion part
The main portion of the text of the Discussion part is devoted to the description of difference between the caves: "The caves of Karts en Yeso de Sorbas have two very different groups of environments that determine the fauna present and whose species affinity is scarce ..." etc.
As I can understand all the caves can be divided into two groups, and then each group can be divided into two subgroups. I would move this to the Materials and Methods part where these characteristics of caves are "hidden" within their descriptions. This division should be formally exposed, as about:
Group 1 ...
- subgroup ... (name of the cave(s))
- subgroup ... (name of the cave(s))
Group 2 ...
- subgroup ... (name of the cave(s))
- subgroup ... (name of the cave(s))
Answer 12. We are not going to follow the suggestion completely. We did think it would be a good idea to present the caves grouped by their characteristics in relation to water, and to describe these characteristics a little more. But this information is used in the discussion, and we are not moving the whole paragraph.
Comment 13. The discussion is about the distribution of cave species in the studied cave. The distribution should be based on the results. I have expected to see the results in the Table 1 that would be consistent. Table 1, however, does not show any records of species by the caves (is it possible to add?). So, we have the only phrase for the Discussion part "Most collembolan species reported and two new species described, Pseudosinella sexocellata sp. nov. and Pygmarrhopalites ruizportaroae sp. nov. seem not to have preference by any of the different ecological types of caves in the karts (!!),
Answer 13. The expression has been rewritten, thanks for the suggestion; now “ecological types of the studied caves”
Comment 14. but Pygmarrhopalites torresi sp. nov. shows preference by those caves without constant water regime.". Can author be not so laconic? A little information about other species would be appropriate. So, the discussion is too short, IMHO. Could you expand it a little pls.
Answer 14. Done. The table has been expanded to include information on how many specimens of each species have been found in each cave. We hope that this is fine with the reviewer. On the other hand, in the Discussion, some information has been included on the fauna in the different caves, considering the different presence of the species in them.
Comment 15. See also the "karts" in two places. Is it OK? Karst?
Answer 15. Sorry, probably the Word spell checker played a trick on us. Karts has been corrected by Karst.
Comment 16. Comments on the Quality of English Language
Considering the evaluation of Quality of English Language. I did not correct English since I am not a native English speaker. It is required to choose, however. I have chosen the first option but my choice does not make much sense.
Submission Date
19 February 2025
Date of this review
06 Mar 2025 11:33:22
Reviewer 2 Report
Comments and Suggestions for Authors
The manuscript „Collembola from the Gypsum Karst of Sorbas (Almería, Spain), with descriptions of three new species” presents a comprehensive work based on a long-term cave arthropoda monitoring and its Collembola faunistic result. The paper presents three new species descriptions from the genera Pygmarrhopalites and Pseudosinella, as well as revision and partial redescriptions of two species from the genera Pseudosinella and Troglopedetes, respectively. The species descriptions the authors provided here are accurate, accompanied by high-quality drawings and SEM photos.
Some minor suggestions:
Pseudosinella sexocellata sp. nov.
I suggest to include an additional drawing of the labial triangle to see the degree of ciliation of labial chaetae M2, E, L1 and L2. It’s an important distinguishing character, and the note „apparently smooth” (Line 265) can be misleading, even if you mention the weak ciliation after.
Pseudosinella najtae
Line 375 – It would be important to know, whether the specimens with 2+2 eyes were found separately, at a certain location, or they occur together with eyeless specimens. In the latter case, it would also be good to know the percentage distribution of individuals without eyes and individuals with 2+2 eyes in a given population. Could there be a correlation between the number of eyes and the parameters of the cave (depth, light, humidity)?
Line 376 – You write that some specimens of the Yesos's caves lacks the Abd IV m chaeta above T2. Is it missing at both side or assimetrically?
For minor corrections and suggestions, see the attached pdf version.

Author Response
Comment 1. The manuscript “Collembola from the Gypsum Karst of Sorbas (Almería, Spain), with descriptions of three new species” presents a comprehensive work based on a long-term cave arthropoda monitoring and its Collembola faunistic result. The paper presents three new species descriptions from the genera Pygmarrhopalites and Pseudosinella, as well as revision and partial redescriptions of two species from the genera Pseudosinella and Troglopedetes, respectively. The species descriptions the authors provided here are accurate, accompanied by high-quality drawings and SEM photos.
Some minor suggestions:
Pseudosinella sexocellata sp. nov.
I suggest to include an additional drawing of the labial triangle to see the degree of ciliation of labial chaetae M2, E, L1 and L2. It’s an important distinguishing character, and the note „apparently smooth” (Line 265) can be misleading, even if you mention the weak ciliation after.
Answer 1. Done; the additional drawing is a group of three chaetae (x3): two of the labial final row (M1 and M2) and one of the post labial, where there are differences too.
Pseudosinella najtae
Comment 2. Line 375 – It would be important to know, whether the specimens with 2+2 eyes were found separately, at a certain location, or they occur together with eyeless specimens. In the latter case, it would also be good to know the percentage distribution of individuals without eyes and individuals with 2+2 eyes in a given population. Could there be a correlation between the number of eyes and the parameters of the cave (depth, light, humidity)?
Answer 2. There is no correlation between the presence of eyed specimens and the caves in which they appear. The proportion of eyed specimens is low, estimated at 10–20 %, and given the large number of specimens studied (1140), it was indeed a stroke of luck to realise that they existed (only a small proportion of them, around 10 %, were mounted). In any case, the suggestion has been taken into account and this information has been added to the paragraph regarding this.
Comment 3. Line 376 – You write that some specimens of the Yesos's caves lacks the Abd IV m chaeta above T2. Is it missing at both side or assimetrically?
Answer 3. Rewritten: “, sometimes at both sides, sometimes asymmetrically” has been added (now in line 431).
Comment 4. For minor corrections and suggestions, see the attached pdf version.
Answer 4. All considered.
Reviewer 3 Report
Comments and Suggestions for Authors
The manuscript provides valuable insights into the Collembola fauna of Spanish gypsum caves. The figures are well-prepared, particularly the SEM photos. The Simple Summary could be slightly condensed, and certain details from the Abstract might be better placed in the Materials and Methods section. While the species descriptions are generally well-done, the diagnoses of both Pygmarrhopalites species could be expanded to include additional key taxonomic features. Furthermore, several corrections are necessary, as noted in the remarks below and within the manuscript text. As for Pseudosinella, I hope other reviewers, Entomobryidae experts, will analyze in more details. Once these corrections are made, the manuscript is recommended for publication in Insects.
Lines 13-29. Possibly, Simple Summary can be somewhat shortened.
Line 17. Instead of “multiplies families” better to give the exact number of families.
Line 18. See remark in the text.
Lines 37-43. This passage is more suited to the Materials and Methods section than to the Abstract
Lines 96-139. Is it possible to add temperatures inside caves when describing them?
Line 137. See remark in the text.
Line 189. Add reference for labial palp nomenclature: (Fjellberg 1999).
Table 1. Pygmarrhopalites pygmaeus (not pigmaeus)
Line 222. Name of the new species Pseudosinella sexocellata : of course, it is authors choice, but seems too similar with already existing P. sexoculata - potential for confusion.
Lines 248-250. See remark in the text.
Line 433. Remove.
Figure 8A. Seta should be included into the row f – see remark.
Line 482-485. Probably the Diagnosis can be expended with including some more taxonomically valuable characters. Also see remarks in the text.
Line 490. “head, 0.7 mm (0.6–0.8 mm); body 0.8 mm”. Head almost as long as body? Looks not realistic.
Line 492. “d: 8+8” but 7 in Fig. 8A
Line 493. “and another one between rows e and f”. This chaeta should be included into row f as it was shown in many other species descriptions.
Line 498. See remark in the text.
Line 505. “Ant II with 16 chaetae”. 17 in Fig. 9A. (but normally 15 in Pygmarrhopalites)
Line 507. See remark in the text.
Line 515. “FP chaetae (e, ae, pe) and some secondary chaetae”. This is very unusual. Normally only one secondary seta FSa present in region F in Pygmarrhopalites. Please, check in other specimens.
Line 521. “Tibiotarsus with 44 chaetae”. 42 in Fig. 10B, also see remarks in the text.
Line 527. “Tibiotarsus with 44”. 43 in Fig. 10C, also see remarks in text.
Figure 11B. Some setae in dorsal valve not shown in comparison with Fig. 11C.
Line 544. In addition, ratio posterior/anterior dorsal body chaetae is interesting.
Line 545-546. “and any or the other circumanal chaetae broadened, winged or serrated”. In Fig. 11 circumanal chaetae are broadened, some of them lamellate and some bearing single tooth.
Line 553: “manubrium with 4 + 4 posterior chaetae”. Actually, 6 per side according to Fig. 11A
Line 557. Add information about mucro tip (e.g. pointed or narrowed)
Figure 13A. Some chaetae labeling should be switched places (Ia-Iae, IIa-IIae), also see remarks.
Line 562. “There are four two previously described species that share...”. Actually, there are more species with such set of characters, even + with only 1 spine on dens, e.g. European P. kristiani, P. kaprusi, P. tauricus etc.
Line 591-594. May be some other diagnostic characters can be added. Also see remarks.
Figure 14A. Chaeta between rows e and f should be included into row f.
Line 600. “row a: 2 + 2 and an axial chaeta”. In Fig. 14A row a is 4+1axial+4.
Line 601-602. “an additional chaeta between rows e and f”. This chaeta should be included into row f as it shown in many other species descriptions. Thus, row f: 6+6 chaetae.
Line 605. See remark in the text.
Line 609. Please, add relative ratios as in previous species: 1/x/y/z
Line 610. Add number of subsegments in Ant IV.
Line 611. Add number of setae in Ant II and III.
Line 614. There are no figures of legs, so please at least shortly describe: number of setae in trochanters, femora, tibiotarsi; presence of inner tooth in claws, corner tooth in empodia, relative length of apical filaments.
Line 618. “m (m1 thickened)”. Does not look thickened in Fig. 15.
Line 625-626. “any or the other circumanal chaetae broadened, winged or serrated”. Some of them look broadened and some with teeth in Fig. 15. Besides, chaetotaxy of dorsal valve seems incomplete.
Line 629. Add number of manubrium setae.
Fig. 16A. Some chaetae labeling should be switched places (Ia-Iae, IIa-IIae), also see remarks.
Line 640-641. “common characters have been omitted”. Nevertheless, some omitted basic characters should be at least briefly described (legs, manubrium etc) to avoid confusion for other researches.
Line 641. “there are four two previously described species that share...”. There are more such species - see remark to previous species.
Abbreviations. See remark in the text.

I am not a native speaker, but it seems that some small corrections could be made.
Author Response
First of all, we would like to thank this reviewer for his comments and suggestions. His detailed analysis of the manuscript, together with his experience with the genre (something that is very evident), have allowed us to detect many errors and have favoured an increase in the quality of the future paper. We are very pleased to have been able to count on this review.
The manuscript provides valuable insights into the Collembola fauna of Spanish gypsum caves. The figures are well-prepared, particularly the SEM photos. The Simple Summary could be slightly condensed, and certain details from the Abstract might be better placed in the Materials and Methods section. While the species descriptions are generally well-done, the diagnoses of both Pygmarrhopalites species could be expanded to include additional key taxonomic features. Furthermore, several corrections are necessary, as noted in the remarks below and within the manuscript text. As for Pseudosinella, I hope other reviewers, Entomobryidae experts, will analyze in more details. Once these corrections are made, the manuscript is recommended for publication in Insects.
Comment 1. Lines 13-29. Possibly, Simple Summary can be somewhat shortened.
Answer 1. Done: from 190 to 110 words.
Comment 2. Line 17. Instead of “multiplies families” better to give the exact number of families.
Answer 2. Done
Comment 3. Line 18. See remark in the text. Lines 37-43. This passage is more suited to the Materials and Methods section than to the Abstract
Answer 3. Done.
Comment 4. Lines 96-139. Is it possible to add temperatures inside caves when describing them?
Answer 4. Done.
Comment 5. Line 137. See remark in the text.
Answer 5. Done.
Comment 6. Line 189. Add reference for labial palp nomenclature: (Fjellberg 1999).
Answer 6. Done. Taking advantage of this comment, we have realized that the Fjellberg 1984 citation was not included in "References" either, so both have been added and the rest renumbered.
Comment 7. Table 1. Pygmarrhopalites pygmaeus (not pigmaeus)
Answer 7. Done.
Comment 8. Line 222. Name of the new species Pseudosinella sexocellata : of course, it is authors choice, but seems too similar with already existing P. sexoculata - potential for confusion.
Answer 8. It is a personal decision of Prof. Jordana
Comment 9. Lines 248-250. See remark in the text.
Answer 9. Done.
Comment 10. Line 433. Remove.
Answer 10.¡Done! ¡Thanks!
Comment 11. Figure 8A. Seta should be included into the row f – see remark.
Answer 11. Done.
Comment 12. Line 482-485. Probably the Diagnosis can be expended with including some more taxonomically valuable characters. Also see remarks in the text.
Answer 12. Done, in part.
Comment 13. Line 490. “head, 0.7 mm (0.6–0.8 mm); body 0.8 mm”. Head almost as long as body? Looks not realistic.
Answer 13. Corrected and rewritten.
Comment 14. Line 492. “d: 8+8” but 7 in Fig. 8A
Answer 14. Done.
Comment 15. Line 493. “and another one between rows e and f”. This chaeta should be included into row f as it was shown in many other species descriptions.
Answer 15. Done.
Comment 16. Line 498. See remark in the text.
Answer 16. Deleted.
Comment 17. Line 505. “Ant II with 16 chaetae”. 17 in Fig. 9A. (but normally 15 in Pygmarrhopalites)
Answer 17. See the answer below.
Comment 18. Line 507. See remark in the text.
Answer 18. Done.
Comment 19. Line 515. “FP chaetae (e, ae, pe) and some secondary chaetae”. This is very unusual. Normally only one secondary seta FSa present in region F in Pygmarrhopalites. Please, check in other specimens.
Line 521. “Tibiotarsus with 44 chaetae”. 42 in Fig. 10B, also see remarks in the text.
Line 527. “Tibiotarsus with 44”. 43 in Fig. 10C, also see remarks in text.
Figure 11B. Some setae in dorsal valve not shown in comparison with Fig. 11C.
Line 544. In addition, ratio posterior/anterior dorsal body chaetae is interesting.
Line 545-546. “and any or the other circumanal chaetae broadened, winged or serrated”. In Fig. 11 circumanal chaetae are broadened, some of them lamellate and some bearing single tooth.
Line 553: “manubrium with 4 + 4 posterior chaetae”. Actually, 6 per side according to Fig. 11A
Line 557. Add information about mucro tip (e.g. pointed or narrowed)
Figure 13A. Some chaetae labeling should be switched places (Ia-Iae, IIa-IIae), also see remarks.
Line 562. “There are four two previously described species that share...”. Actually, there are more species with such set of characters, even + with only 1 spine on dens, e.g. European P. kristiani, P. kaprusi, P. tauricus etc.
Line 591-594. May be some other diagnostic characters can be added. Also see remarks.
Figure 14A. Chaeta between rows e and f should be included into row f.
Line 600. “row a: 2 + 2 and an axial chaeta”. In Fig. 14A row a is 4+1axial+4.
Line 601-602. “an additional chaeta between rows e and f”. This chaeta should be included into row f as it shown in many other species descriptions. Thus, row f: 6+6 chaetae.
Line 605. See remark in the text.
Line 609. Please, add relative ratios as in previous species: 1/x/y/z
Line 610. Add number of subsegments in Ant IV.
Line 611. Add number of setae in Ant II and III.
Line 614. There are no figures of legs, so please at least shortly describe: number of setae in trochanters, femora, tibiotarsi; presence of inner tooth in claws, corner tooth in empodia, relative length of apical filaments.
Line 618. “m (m1 thickened)”. Does not look thickened in Fig. 15.
Line 625-626. “any or the other circumanal chaetae broadened, winged or serrated”. Some of them look broadened and some with teeth in Fig. 15. Besides, chaetotaxy of dorsal valve seems incomplete.
Line 629. Add number of manubrium setae.
Fig. 16A. Some chaetae labeling should be switched places (Ia-Iae, IIa-IIae), also see remarks.
Line 640-641. “common characters have been omitted”. Nevertheless, some omitted basic characters should be at least briefly described (legs, manubrium etc) to avoid confusion for
Line 641. “there are four two previously described species that share...”. There are more such species - see remark to previous species.
Answer 19. See the answer below. Almost all suggestions and comments have been considered.
Comment 20. Abbreviations. See remark in the text.
Answer 20. Done.
---
Answer to the suggestions in the PDF
Simple Summary
Comment 21. “Collembola (springtails)”
Answer 21. has been rewritten as “springtails (Collembola)”
Comment 22. “multiple families”, now “nine families”
Answer 22. Done.
Comment 23. “four newly described”
Answer 23. now “three newly described” (we considered the partial redescription of P. najtae, but it is true that is not a new species).
Abstract
Comment 24. About the paragraph: “Taxonomic and morphological…”
Answer 24. has been deleted since this information, with a more extensive wording, is included in Material and Methods.
Material and Methods, Caves studied
About temperatures…
Nomenclature
Comment 25. Fjellberg (1999) has been added because has been considered for labial palp.
Answer 25. Fjellberg (1999) has been added because has been considered for labial palp..
Comment 26. Table 1, “pygmaeus”.
Answer 26. Done.
Pseudosinella sexocellata sp. nov. description
Comment 27. Pseudosinella sexocellata
Answer 27. yes, but it is a personal decision of Prof. Jordana.
Comment 28. Diagnosis, the word “moderately”
Answer 28. has been deleted.
Comment 29. “Area not ringed”
Answer 29. now “Not ringed area”.
Comment 30. “Claw elongate”
Answer 30. now “Claws elongated”.
Comment 31. About the expression "labral row a"
Answer 31. what we mean is that row a has smooth chaetae.
Comment 32. “projection chaeta like”
Answer 32. now “chaeta-like projection”, thanks.
Comment 33. “distal”
Answer 33. now “anteapical”.
Troglopedetes machadoi
Comment 34. Figure 7,
Answer 34. “dentes”, now “dens”.
Comment 35. About “Subfamily Entomobryinae Schäffer, 1896 [52] sensu Zhang and Deharveng 2015 [35]”,
Answer 35. thanks, it was a big mistake.
Pygmarrhopalites ruizporteorae
Comment 36. Figure 8
Answer 36. redrawn to integrate the orphan point into the f line (same for Figure 14, P. torresi).
Comment 37. Diagnosis
Answer 37. rewritten.
Comment 38. Description, head and body length, and other cuestions
Answer 38. all corrected, thanks. Regarding the number of chaetae from Ant II, the material has been reviewed (as stated in some parts of the paper, which is very damaged by the sampling method) and confirmed that there are 15. Some issues have required re-examining specimens, the number of chaetad on legs and antennae has had to be confirmed, and the corresponding drawings have been changed. In all cases, the reviewer has directed us to information consistent with what is usual in the genus.
Pygmarrhopalites torresi
Comment 39.
Answer 39. Similar to what happened with the previous species, errors have been corrected and all doubts raised have been clarified. There were some errors (for example the description of Th II chaetotaxy).
Comment 40. Fig. 15: “Besides, chaetotaxy of dorsal valve looks incomplete”
Answer 40. yes, the drawing is not large enough to include everything; this part is only indicative. This has been indicated in the figure legend. A new drawing (legs) has been included, and revised the number of chaetae of manubrium and legs.